# BATTERY-SIM-AGENT: LEVERAGING LLM-AGENT FOR INVERSE BATTERY PARAMETER ESTIMATION

## ABSTRACT

Parameterizing high-fidelity "digital twins" of batteries is a critical yet challenging inverse problem that hinders the pace of battery innovation. Prevailing methods formulate this as a black-box optimization (BBO) task, employing algorithms that are sample-inefficient and blind to the underlying physics. In this work, we introduce a new paradigm that reframes the inverse problem as a reasoning task, and present BATTERY-SIM-AGENT, the first framework to deploy a Large Language Model (LLM) agent in a closed loop with a high-fidelity battery simulator. The agent mimics a human scientist's workflow: it interprets rich, multi-modal feedback from the simulator, forms physically-grounded hypotheses to explain discrepancies, and proposes structured parameter updates. On a systematically constructed benchmark suite spanning diverse battery chemistries, operating conditions, and difficulty levels, our agent significantly outperforms strong BBO baselines like Bayesian optimization in identifying accurate parameters. We further demonstrate the framework's capability in complex long-horizon degradation fitting tasks and validate its practical applicability on real-world battery datasets. Our results highlight the promise of LLM-agents as reasoning-based optimizers for scientific discovery and battery parameter estimation.

## 1 INTRODUCTION

The transition to a sustainable energy future is intrinsically linked to advancements in battery technology (Hamdan et al.). From electrifying transportation to stabilizing power grids, next-generation batteries are a critical need (Hamdan et al.). However, the physical development and testing of these batteries is a major bottleneck (Attia et al.; Román-Ramírez & Marco). Characterizing a battery's performance and degradation over its lifetime can require thousands of hours of continuous cycling (Stroe et al., 2018). A promising alternative is to build *digital twins*—high-fidelity virtual replicas instantiated in physics-based simulators such as PYBAMM (Sulzer et al., 2021). Yet, realizing this vision hinges on solving a fundamental *inverse problem*: the simulators require microscopic parameters that cannot be directly measured, while only macroscopic data are available. Accurately identifying these parameters is a long-standing challenge in battery engineering (Subramanian & Braatz, 2013; Prasad et al., 2015; Gopinath et al., 2016).

Traditionally, this inverse problem is formulated as a black-box optimization (BBO) task. As detailed in Table 1, researchers have long employed algorithms like Bayesian optimization (Wang & Jiang, 2023; Jiang et al., 2022) or genetic algorithms (Zhang et al., 2014; Magnor & Sauer, 2016; Blaifi et al., 2016) to iteratively query the simulator and minimize the mismatch between simulated and observed data. While flexible, these methods are inherently *blind*: they treat the simulator as an opaque oracle and lack physical intuition. This often leads to high sample complexity and convergence to implausible local minima.

The limitations of blind search motivate a paradigm shift. With the advent of Large Language Models (LLMs) as powerful *reasoning engines*, a new wave of "agentic science" is emerging, where LLM-powered agents automate complex scientific discovery workflows (Wei et al., 2025). These agents have shown success in solving inverse problems in diverse fields like materials science (Wu et al., 2025) and solid mechanics (Ni & Buehler, 2024). This inspires us to ask a central question: *can the inverse problem of battery parameter estimation be reframed not as a brute-force search, but as a reasoning-driven scientific workflow guided by an LLM-agent?*

We answer this question affirmatively by introducing **Battery-Sim-Agent**, a framework that pioneers the use of an LLM-agent in a *simulator-in-the-loop* configuration to solve the inverse problem in battery science. Our agent acts as an AI scientist: in each iteration, it is presented with rich, multimodal feedback that compares the current simulation against experimental data. This includes not only quantitative error metrics but also visual overlays of voltage curves, allowing it to identify qualitative discrepancies like misaligned plateaus or incorrect slopes. Based on this evidence, the agent formulates a physical hypothesis (e.g., "premature voltage drop suggests an electrolyte transport limitation") and proposes a targeted parameter update in a structured JSON format. To ensure stability and long-term planning, the agent is equipped with a persistent memory of its past actions and their outcomes. We validate this framework through a comprehensive experimental suite spanning diverse battery chemistries, operating conditions, and difficulty levels, demonstrating that our agent consistently achieves 67-95% reduction in curve-matching error compared to traditional black-box optimization baselines. We further showcase the framework's capability in complex long-horizon degradation fitting tasks and validate its practical applicability on real-world battery datasets.

Our main contributions can be summarized as follows:

1. We introduce a novel agentic framework that reframes the battery inverse problem from a blind mathematical search into an interpretable, hypothesis-driven scientific workflow, pioneering the use of a simulator-in-the-loop LLM-agent in this domain.

2. We architect a suite of principled modules specifically designed for this workflow, including a multi-modal feedback system that translates complex simulation data into actionable insights for the agent, and a persistent memory to enable robust, long-horizon reasoning.

3. We provide a comprehensive experimental validation of our framework, demonstrating on extensive simulated benchmarks spanning diverse chemistries and difficulty levels, as well as real-world battery datasets, that our reasoning-based approach achieves 67-95% reduction in parameter estimation error compared to traditional black-box optimization methods.

| Aspect | Traditional BBO | Battery-Sim-Agent (Ours) |
|---|---|---|
| Search Paradigm | Blind Search | Hypothesis-Driven |
| Feedback Signal | Scalar Loss | Rich & Multi-modal |
| Interpretability | Low | High |
| Efficiency | Sample-Inefficient | Guided & Efficient |

Table 1: Comparison of Traditional Black-Box Optimization and Battery-Sim-Agent.

## 2 BACKGROUND

### 2.1 THE CHALLENGE OF PARAMETERIZING BATTERY DIGITAL TWINS

A central goal in battery science is to create high-fidelity "digital twins" that can accurately predict a battery's performance and long-term degradation. This is a critical yet challenging task. The degradation of a battery is a slow process, often requiring hundreds or thousands of charge-discharge cycles to observe significant capacity fade. While macroscopic data from these cycles—such as terminal voltage, current, and total capacity—are readily available, they are merely symptoms of underlying microscopic processes.

The true drivers of battery behavior are a set of internal, microscopic physical and chemical parameters. These include properties like the porosity of the electrodes, the diffusion coefficients of lithium ions in the solid and electrolyte phases, and kinetic reaction rates. These parameters, collectively denoted as a vector $\theta$, govern the complex system of coupled partial differential equations (PDEs) that form the core of high-fidelity electrochemical models like the Doyle-Fuller-Newman (DFN) model (Subramanian & Braatz, 2013). However, directly measuring these parameters is often prohibitively expensive, requires specialized laboratory equipment, or is even physically impossible without destroying the battery cell. This creates a fundamental gap between what we can easily observe (macroscopic data) and what we need to know to build an accurate model (microscopic parameters). The task of bridging this gap of inferring the hidden parameters $\theta$ from observable data is known as the inverse problem of parameter estimation in battery science (Prasad et al., 2015; Gopinath et al., 2016).

## 2.2 FORMULATION AS A BLACK-BOX OPTIMIZATION PROBLEM

Traditionally, the inverse problem is formulated as a black-box optimization (BBO) task. The goal is to find a parameter vector $\theta^\star$ that minimizes a loss function, $\mathcal{L}(\theta)$, which quantifies the discrepancy between the simulator's outputs and the experimentally observed data. To overcome the ill-posedness of the problem, this matching must be performed across a set of diverse experimental protocols $\mathcal{P}$ (e.g., different charge/discharge C-rates (Balog & Davoudi, 2013; Pantoja et al., 2022)).

For each protocol $p \in \mathcal{P}$, we collect a set of observed macroscopic trajectories, $Y_p^{\text{obs}}$, which can include terminal voltage $V(t)$, current $I(t)$, and cycle capacity $Q$. The simulator, given parameters $\theta$, produces corresponding simulated trajectories $Y_p^{\text{sim}}(\theta)$. The overall objective is to minimize a composite loss function, typically a weighted sum over all protocols:

$$\theta^\star = \arg\min_\theta \mathcal{L}(\theta), \quad \text{where} \quad \mathcal{L}(\theta) = \sum_{p \in \mathcal{P}} w_p \cdot d\big(Y_p^{\text{sim}}(\theta), Y_p^{\text{obs}}\big) + \lambda R(\theta). \tag{1}$$

Here, $d$ is a distance metric that can compare multiple trajectories, $w_p$ are weights for each protocol used to balance different scales, and $R(\theta)$ is a regularization term. This optimization is notoriously difficult for three main reasons:

- **Expensive, Non-Differentiable Black-Box:** Each evaluation of $\mathcal{L}(\theta)$ requires a full, computationally costly simulation, and the gradients $\nabla_\theta \mathcal{L}$ are typically unavailable.

- **Ill-Posedness:** The problem is ill-posed, meaning many different parameter sets $\theta$ can produce nearly identical output trajectories (a phenomenon known as equifinality), making the minimum of the loss landscape difficult to identify uniquely.

- **High Dimensionality:** The parameter vector $\theta$ can be high-dimensional, making a brute-force search of the parameter space intractable.

## 2.3 SIMULATOR-IN-THE-LOOP AND AGENTIC SCIENCE

The limitations of treating the simulator as an opaque oracle have motivated a shift towards more interactive paradigms. A common approach in computational science is the "simulator-in-the-loop" model, where a human expert iteratively adjusts parameters based on simulation outputs. Recently, the rise of Large Language Models (LLMs) as powerful reasoning engines has opened the door to automating this process at scale (Hu et al., 2025). This has led to the emergence of "agentic science" where LLM agents take on the role of the human scientist (Wei et al., 2025). These agents have shown success across diverse domains: molecular design (Wu et al., 2025), inverse problems in solid mechanics (Ni & Buehler, 2024), and galaxy observation interpretation (Sun et al., 2024). Instead of being guided by a single scalar loss value, LLM agents can interpret rich, structured feedback from simulators—including full data trajectories, visual plots, and diagnostic error messages. This allows agents to reason about physical causes of discrepancies and formulate targeted hypotheses, reframing optimization from a blind search into an intelligent, hypothesis-driven workflow. This emerging paradigm provides the direct motivation for our work.

# 3 METHOD

To address the complex, multi-objective, and heterogeneous optimization challenge formulated in Sec. 2.2, we introduce BATTERY-SIM-AGENT. The core innovation of our framework is to replace the conventional "blind" numerical search of traditional BBO with a reasoning engine that can interpret and act upon the rich, structured information produced by a physics-based simulator. An LLM-agent, acting as an AI scientist, can handle the multi-objective nature of the problem by reasoning about qualitative trade-offs, and navigate the heterogeneous parameter space by proposing targeted, mechanism-aware updates. This allows us to reframe the inverse problem as an interpretable, hypothesis-driven workflow.

## 3.1 AGENT-DRIVEN OPTIMIZATION FORMULATION

Aligned with the optimization objective formulated in Eq. equation 1, our overall goal is to find parameters $\theta^\star$ that minimize the composite loss $\mathcal{L}(\theta)$. However, unlike traditional methods that

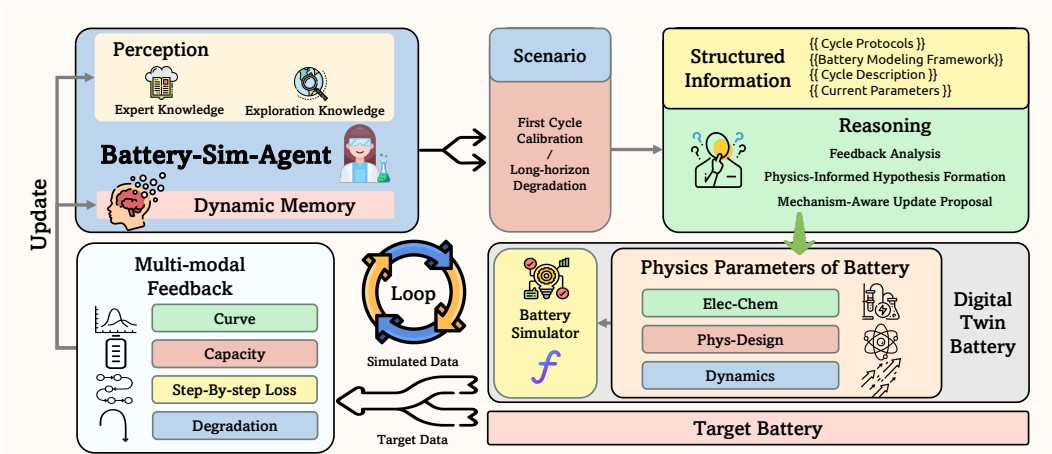

Figure 1: The closed-loop workflow of BATTERY-SIM-AGENT. The agent proposes parameters for the PYBAMM simulator. The simulator's output is then compared against target data to generate structured, multi-modal feedback (Sec. 3.2), which the agent analyzes using its dynamic memory (Sec. 3.3) to reason about the next parameter update.

aggregate multiple objectives into a single scalar, our agent operates on a disaggregated set of objectives. The target is not a single value, but a set of discrepancies across various physical quantities:

$$\mathcal{L}(\theta) = \{d_V(V_{\text{sim}}, V_{\text{obs}}), d_Q(Q_{\text{sim}}, Q_{\text{obs}}), \dots\}. \tag{2}$$

The regularization $R(\theta)$ is also enforced implicitly by the agent's reasoning, guided by the physical priors stored in its memory $\mathcal{M}_t$. The agent-driven framework bypasses manual loss weighting by receiving a structured feedback signal $F_t$ containing the individual discrepancy components and proposing an update $\Delta\theta_t = \Phi_{\text{LLM}}(F_t, \mathcal{M}_t)$ to jointly improve the objectives. The agent function $\Phi_{\text{LLM}}$ is realized by querying a Large Language Model with a structured prompt that encapsulates the feedback $F_t$ and relevant knowledge from memory $\mathcal{M}_t$. The iterative update rule is:

$$\theta_{t+1} = \Pi_{[\ell,u]}(\theta_t + \eta_t \Delta\theta_t), \tag{3}$$

where $\Pi$ is a projection to enforce physical bounds and $\eta_t$ is an adaptive step size.

By disaggregating the objective, we enable the agent to perform *causal attribution*. The LLM can map specific feature mismatches (symptoms) to specific parameter subsets (causes), effectively navigating the high-dimensional parameter space by decomposing the problem into physically meaningful sub-problems.

### 3.2 THE HYPOTHESIS-DRIVEN REASONING LOOP

The agent's workflow mimics a human scientist, proceeding in three steps within each iteration. Direct mapping from observation to parameters using LLMs can lead to hallucinations, therefore, we design a structured reasoning loop that enforces a Chain-of-Thought (CoT) process.

**Step 1: Analyze Feedback.** The agent receives a multi-modal feedback package $F_t$ in a structured JSON format. This contains not just overall error metrics, but also fine-grained, feature-space residuals that a human expert would examine:

```
{
    "residuals": { "capacity_mape": 0.08, "voltage_rmse": 0.05 },
    "features": { "cc_charge_time_mismatch_s": -120.5, "plateau_shift_v": -0.02 },
    "visual": "path/to/voltage_curve_overlay.png",
    "events": ["simulation_success"]
}
```

This translation bridges the modality gap. While LLMs struggle to interpret raw floating-point arrays, they excel at reasoning with semantic descriptions of trends and shapes.

**Step 2: Reason and Hypothesize.** Guided by its memory $\mathcal{M}_t$, the agent analyzes this rich feedback to form a causal hypothesis. The prompt encourages a scientific reasoning process:

```
"Given the feedback, especially the short CC charge time and the low
 voltage plateau, what is the most likely physical cause? Formulate a
 hypothesis and decide on a corrective strategy."
```

This intermediate reasoning step serves as a "cognitive check." By forcing an explicit hypothesis, we ground the agent's actions in physical laws, reducing the likelihood of proposing physically implausible parameters.

**Step 3: Propose a Structured Update.** Finally, the agent is prompted to translate its hypothesis into a concrete, machine-actionable update, which it returns in a strict JSON format, ensuring reliability and interpretability:

```
"Based on your hypothesis, propose a targeted parameter update:"
{
  "updated_params": { "Positive electrode reaction rate [s^-1]": "*1.2" },
  "rationale": "Increasing the positive reaction rate by 20% should
                raise the voltage plateau and extend the CC charge time."
}
```

BATTERY-SIM-AGENT performs targeted local adjustments based on the specific hypothesis derived in the previous step, rather than relying on global exploration of the parameter space.

### 3.3 DYNAMIC MEMORY WITH KNOWLEDGE WARM-UP

The agent's ability to reason effectively relies on its memory, $\mathcal{M}_t$, which dynamically incorporates both expert knowledge and empirical findings.

**Initial Knowledge Injection.** We initialize the memory $\mathcal{M}_0$ with human expert knowledge from the literature and our own domain expertise. This includes fundamental parameter information (e.g., physical bounds) and a set of fuzzy, qualitative rules-of-thumb.

**Trial-and-Error Warm-up Phase.** Before the main optimization loop, the agent undergoes a "warm-up" phase to build a preliminary causal model of parameter effects. It generates random perturbations around $\theta_0$ and executes simulations. The resulting feedback is *not* for optimization, but is processed by the LLM to enrich its memory. The agent is prompted to summarize the outcomes into learned sensitivity rules (e.g., "Observed: perturbing 'Negative electrode thickness' by +10% strongly increases capacity but causes simulation failure at high C-rates"). Since we cannot compute the gradient $\nabla_\theta \mathcal{L}$ directly, this phase effectively allows the agent to build an "internal mental model" of the local sensitivity landscape. This learned knowledge makes the subsequent optimization search significantly more targeted and robust.

### 3.4 INSTANTIATED PIPELINES FOR KEY SCIENTIFIC SCENARIOS

The following two pipelines showcase the flexibility of our framework in tackling both a short-horizon, high-fidelity matching task and a long-horizon, dynamic tracking task.

**First-Cycle Calibration.** This scenario focuses on matching the detailed voltage curve of the initial cycles. It relies heavily on multi-modal feedback and the agent's ability to perform protocol-aware staged matching. For a standard CC-CV protocol, the agent is prompted to analyze the CC and CV phases separately, attributing mismatches to different physical phenomena (e.g., kinetics vs. transport limitations), a nuanced strategy that is difficult to encode in a simple loss function.

**Long-Horizon Degradation Fitting.** This scenario aims to capture capacity fade over hundreds of cycles by fitting SEI-related degradation parameters. To handle the vast amount of data, we employ a **dynamic cycle indexing** mechanism. Instead of analyzing all cycles, the agent is shown the full

---

**Algorithm 1** The Two-Phase Workflow of BATTERY-SIM-AGENT

---

1: **Input:** Target data $Y^{\text{obs}}$, parameter bounds $[\ell, u]$, budget $T$, warm-up steps $N_w$
2: Initialize memory $\mathcal{M}_0$ with human knowledge
3: // **Phase 1: Trial-and-Error Warm-up**
4: **for** $k = 1$ to $N_w$ **do**
5: $\quad$ Generate a random perturbation $\delta_k$ around $\theta_0$
6: $\quad$ $Y^{\text{sim}} \leftarrow \text{SIMULATE}(\theta_0 + \delta_k)$
7: $\quad$ $F_k \leftarrow \text{BUILDFEEDBACK}(Y^{\text{sim}}, Y^{\text{obs}})$
8: $\quad$ $\mathcal{M}_k \leftarrow \text{UPDATEMEMORY}(\mathcal{M}_{k-1}, F_k, \text{"Summarize sensitivity"})$
9: **end for**
10: // **Phase 2: Main Optimization Loop**
11: **for** $t = 0$ to $T - 1$ **do**
12: $\quad$ $Y^{\text{sim}} \leftarrow \text{SIMULATE}(\theta_t)$
13: $\quad$ $F_t \leftarrow \text{BUILDFEEDBACK}(Y^{\text{sim}}, Y^{\text{obs}})$
14: $\quad$ $\Delta\theta_t, \text{rationale}_t \leftarrow \text{QUERYLLM}(F_t, \mathcal{M}_{N_w+t-1})$
15: $\quad$ $\theta_{t+1} \leftarrow \Pi_{[\ell,u]}\big(\theta_t + \eta_t\, \Delta\theta_t\big)$
16: $\quad$ $\mathcal{M}_{N_w+t} \leftarrow \text{UPDATEMEMORY}(\mathcal{M}_{N_w+t-1}, F_t, \Delta\theta_t, \text{rationale}_t)$
17: $\quad$ **if** converged **then**
18: $\quad\quad$ **break**
19: $\quad$ **end if**
20: **end for**
21: **return** $\theta_{t^\star}$

---

degradation curve and is prompted to select a small, informative subset of cycle indices (e.g., start, end, points of maximum curvature) for detailed feedback. This ensures the feedback is both compact and highly relevant for capturing the long-term degradation dynamics.

# 4 RELATED WORK

Our work is positioned at the intersection of battery science and the emerging field of AI-driven scientific discovery. The inverse problem of identifying microscopic parameters for high-fidelity electrochemical models, such as the Doyle-Fuller-Newman (DFN) model implemented in simulators like PYBAMM, is a long-standing challenge in battery engineering (Sulzer et al., 2021; Subramanian & Braatz, 2013). The problem is notoriously ill-posed, with many parameter combinations yielding similar macroscopic outputs (Gopinath et al., 2016; Prasad et al., 2015). Historically, this challenge has been addressed using classical black-box optimization (BBO) methods, such as Bayesian optimization or evolutionary algorithms (Wang & Jiang, 2023; Jiang et al., 2022; Zhang et al., 2014). While versatile, these methods are fundamentally "blind" optimizers; they treat the simulator as an opaque oracle and lack physical intuition, often resulting in high sample complexity and convergence to implausible solutions.

Concurrently, a paradigm shift is underway in how AI is applied to science, moving from data analysis to autonomous discovery. Large Language Models (LLMs) are increasingly used as "cognitive partners" for tasks like hypothesis generation and literature synthesis (Zuo et al., 2025; Hu et al., 2025). More powerfully, they are being deployed as the core reasoning engine in autonomous agents that can interact with external tools in a closed loop, a trend often referred to as "agentic science" (Wei et al., 2025). This agent-based approach has already shown significant promise in solving complex parameter tuning and design problems in diverse scientific and engineering domains, such as materials science (Wu et al., 2025), solid mechanics (Ni & Buehler, 2024), astrophysics (Sun et al., 2024), and hyperparameter optimization (Liu et al., 2024).

Human-AI collaborative optimization frameworks further integrate expert knowledge into the search loop. COBOL (Xu et al.) augments Bayesian Optimization with human "accept/reject" feedback and provides theoretical no-harm and handover guarantees. In contrast, our Battery-Sim-Agent treats the LLM not as a verifier but as a generative reasoner proposing continuous parameter updates. While this offers richer semantic guidance grounded in physical intuition, it lacks the formal regret bounds available in COBOL—an opportunity for future hybrid approaches.

Most relevant to our work is the emerging use of LLMs for parameter inference in physical systems. SimLM (Memery et al.) demonstrated simulator-in-the-loop reasoning on simple kinematic problems, though performance degraded on slightly more complex dynamics. Our work extends this paradigm to high-fidelity engineering systems: battery parameter estimation requires navigating coupled PDEs, high-dimensional parameter spaces, and pronounced equifinality—far beyond the low-dimensional settings addressed in prior work.

These works demonstrate the potential of LLM-agents to navigate complex search spaces more intelligently than traditional algorithms. Building upon these foundations, our work is the first to bridge these two domains. We introduce an LLM-agent as a reasoning-based optimizer to specifically tackle the challenging inverse problem in battery science.

## 5 EXPERIMENTS

We conduct comprehensive experiments on simulated benchmarks and real-world data to evaluate BATTERY-SIM-AGENT. Our evaluation demonstrates the superiority of the reasoning-based approach across diverse battery chemistries, operating conditions, and difficulty levels. Specifically, our experimental design explores a new paradigm for addressing the inverse problem of battery parameter estimation through physics-grounded reasoning. We hypothesize that an agent capable of forming and testing causal hypotheses can navigate the parameter landscape with greater efficiency and robustness. To assess this, we structure our experiments across three progressively challenging tiers: (1) **Controlled Benchmarks** to rigorously quantify parameter accuracy against ground truth; (2) **Complex Dynamics** involving long-horizon degradation to test reasoning over time; and (3) **Practical Validation** on real-world data to evaluate applicability in noisy, uncertain environments.

### 5.1 EXPERIMENTAL SETUP

**Benchmark Test Suite.** We construct a diverse benchmark suite using the high-fidelity Doyle-Fuller-Newman (DFN) model (Doyle et al., 1993) in PYBAMM (Sulzer et al., 2021). To address the challenge of defining a consistent evaluation metric across heterogeneous systems, our construction follows a strict "Base-Perturbation-Filter" pipeline. This ensures that every task represents a realistic inverse problem where the agent starts with a known prior ($\theta_{\text{init}}$) and must recover an unknown ground truth ($\theta^*$):

**1. Base Chemistries (The Priors):** We employ five classic, well-established parameter sets from the literature: Chen2020 (Chen et al., 2020) (NMC811/graphite), ORegan2022 (O'Regan et al., 2022) (NMC532/graphite), Prada2013 (Prada et al., 2013) (LFP/graphite), Ecker2015 (Ecker et al., 2015b;a) (NMC111/graphite), and Marquis2019 (Marquis et al., 2019) (NMC622/graphite). These serve as the initial parameter guess $\theta_{\text{init}}$ for the agent in each task.

**2. Target Generation (The Ground Truth):** To generate the "unknown" target data $Y_{\text{obs}}$, we apply controlled perturbations to the base parameters to create a ground truth vector $\theta^*$. We define two difficulty modes:

- **Regular Mode (Multi-Parameter):** We apply 12 expert-designed, physically-plausible multi-parameter perturbations that represent realistic manufacturing variations or design choices (e.g., simultaneously altering electrode thickness and porosity). These combinations are carefully crafted to maintain physical plausibility while creating meaningful optimization challenges.

- **Extreme Mode (Single-Parameter):** We apply large perturbations ($0.5\times$ to $2.0\times$) to one of 9 key parameters (particle radiation, electrode thicknesses, porosities, Bruggeman coefficients, separator thickness), creating challenging cases that often push the simulator to its stability limits.

**3. Varied Operating Conditions:** For each chemistry, we generate ground-truth data under three different charge/discharge protocols (0.2C, 1C, and 2C), simulating a range of operational severities from gentle to aggressive cycling conditions.

**Data Generation and Filtering Process.** Our systematic data generation follows a rigorous multi-stage process. We iterate through all combinations of base parameter sets, C-rates, and perturbation

rules, then apply a two-stage filtering process: (1) We discard parameter combinations that result in simulation failures in PYBAMM, ensuring numerical stability; (2) We filter out cases where the resulting capacity change is less than 1% compared to baseline, ensuring each test case presents a meaningful, non-trivial challenge. This process results in 233 valid combinations for extreme mode and 373 for regular mode, from which we randomly sample 100 cases each to form our final evaluation suite of 200 unique tasks. In each task, the agent is initialized at $\theta_{\text{init}}$ and must recover the hidden $\theta^*$ by minimizing the discrepancy with $Y_{\text{obs}}$. Detailed generation rules and examples are provided in Appendix B.

**Baselines and Comparison Strategy.** We compare our full agent against strong baselines and an ablation to isolate the benefits of different components:

- **Battery-Sim-Agent-O3:** The full agent powered by GPT-O3 (OpenAI, 2025), incorporating our complete reasoning workflow with hypothesis generation, iterative refinement, and multi-objective optimization capabilities.

- **Battery-Sim-Agent-OSS:** An ablation using GPT-OSS (OpenAI et al., 2025), a powerful 120B parameter open-source model, but without the chain-of-thought reasoning capabilities of our full agent. This isolates the benefit of the reasoning workflow itself.

- **Bayesian Optimization (BO):** We use standard Bayesian Optimization implemented by Meta's Ax platform (Olson et al., 2025), representing state-of-the-art black-box optimization methods commonly used in parameter estimation.

We also experimented with other evolutionary algorithms including CMA-ES (Hansen et al., 2019), but found that these methods generally failed to converge on our challenging parameter estimation tasks. We also present results of **Default Parameters**, which includes the original parameter values from each literature source as a naive baseline, representing the performance when using published parameters without optimization.

**Evaluation Metrics.** We evaluate performance using comprehensive error metrics between predicted and ground-truth voltage/capacity curves: Mean Absolute Percentage Error (MAPE) and Root Mean Squared Error (RMSE). These metrics capture both relative and absolute deviations, providing a thorough assessment of parameter identification accuracy.

## 5.2 RESULTS ON FIRST-CYCLE CALIBRATION

Figure 2 and Table 2 present our comprehensive results for first-cycle calibration. The findings clearly demonstrate the superiority of our reasoning-based approach across all evaluation scenarios. Specifically, **Battery-Sim-Agent-O3** consistently and significantly outperforms all other methods across both regular and extreme modes. As shown in Fig. 2, our agent achieves not only substantially lower median error but also dramatically reduced variance, indicating more reliable and stable performance. The ablation (OSS) performs better than BO methods but is clearly inferior to our full agent, confirming that the agent's explicit reasoning capabilities are critical to its success.

The quantitative results in Table 2 reveal the magnitude of our improvements. In regular mode, our agent achieves MAPE reductions of 67-95% compared to BO across different chemistries, with particularly impressive performance on Ecker2015 (0.77% vs 27.37% MAPE) and Marquis2019 (1.27% vs 13.54% MAPE). The ablation study demonstrates that while GPT-OSS provides some benefit over traditional optimization, our full reasoning workflow delivers substantial additional improvements. In **extreme mode**, where single parameters are dramatically perturbed, the performance of baseline optimizers degrades significantly due to the highly non-convex optimization landscape. In contrast, our agent's reasoning capabilities allow it to maintain robust performance by systematically exploring the parameter space and adapting its search strategy based on intermediate results.

**C-rate Performance Analysis.** Figure 3 shows performance across different charge/discharge protocols. Our agent maintains superior performance across all C-rates, with particularly notable improvements at higher rates where traditional optimization methods struggle with the increased complexity of the electrochemical dynamics.

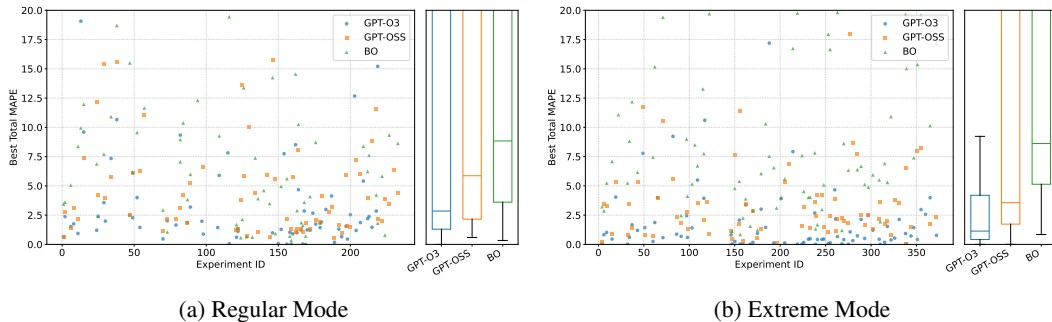

(a) Regular Mode          (b) Extreme Mode

Figure 2: **Main results on first-cycle calibration.** Our reasoning-based agent (GPT-O3) consistently outperforms its ablation (GPT-OSS) and Bayesian Optimization across both difficulty modes, achieving lower median error and significantly reduced variance.

Table 2: Detailed MAPE and RMSE results for first-cycle calibration across modes and chemistries.

| Mode | Methods | Chen2020 | ORegan2022 | Ecker2015 | Prada2013 | Marquis2019 |
|------|---------|----------|-----------|-----------|-----------|-------------|
| | | **MAPE** | | | | |
| Regular | Default | 159.09±118.6 | 160.47±119.0 | 108.42±65.1 | 79.16±56.1 | 122.60±79.2 |
| | BO | 211.97±404.4 | 81.73±224.0 | 27.37±80.3 | 56.31±222.2 | 13.54±9.3 |
| | BatterySimAgent-OSS | 38.05±100.0 | 46.34±53.3 | 7.63±17.3 | 23.74±44.7 | 9.55±20.3 |
| | BatterySimAgent-O3 | **12.60±24.5** | **34.18±48.2** | **0.77±1.2** | **5.97±12.2** | **1.27±1.1** |
| Extreme | Default | 119.32±110.4 | 181.95±181.2 | 100.43±115.3 | 150.65±87.9 | 108.00±89.7 |
| | BO | 159.41±362.4 | 84.55±213.7 | 137.31±278.6 | **17.05±25.3** | **8.42±6.3** |
| | BatterySimAgent-OSS | 23.66±42.2 | 50.88±61.5 | 45.47±92.2 | 23.96±42.3 | 19.50±39.1 |
| | BatterySimAgent-O3 | **23.38±52.4** | **19.44±24.2** | 27.85±79.5 | 59.14±62.3 | 48.34±90.4 |
| | | **RMSE** | | | | |
| Regular | Default | 5.47±2.7 | 6.47±3.0 | 1.30±0.4 | 2.68±1.5 | 1.64±0.5 |
| | BO | 2.50±3.4 | **2.27±1.7** | 0.21±0.1 | 0.57±0.4 | 0.26±0.1 |
| | BatterySimAgent-OSS | 1.87±2.5 | 3.07±2.6 | 0.26±0.2 | 1.22±1.5 | 0.43±0.4 |
| | BatterySimAgent-O3 | **1.18±1.9** | 2.41±2.4 | **0.06±0.1** | **0.32±0.4** | **0.19±0.1** |
| Extreme | Default | 4.23±2.3 | 6.11±3.0 | 1.74±1.2 | 5.21±2.8 | 1.48±0.8 |
| | BO | 1.63±2.9 | 2.10±2.1 | 0.77±2.5 | **0.60±0.5** | **0.26±0.2** |
| | BatterySimAgent-OSS | 1.91±2.2 | 3.04±2.8 | 0.69±1.0 | 1.23±1.3 | 0.47±0.5 |
| | BatterySimAgent-O3 | **1.48±2.6** | **1.53±1.5** | **0.43±0.8** | 2.50±2.3 | 0.59±0.8 |

## 5.3 Advanced Applications

**Long-Horizon Degradation Fitting.** We extend our evaluation to degradation scenarios requiring simultaneous fitting of electrochemical and SEI parameters, representing a significantly more challenging optimization problem. Table 3 demonstrates that BATTERY-SIM-AGENT framework successfully handles this complex task across both model variants. Interestingly, BatterySimAgent-OSS achieves superior performance in degradation fitting (1.37% vs 1.77% Total MAPE), suggesting that the reasoning complexity should match task characteristics, for smooth, long-horizon degradation trends, OSS's more direct optimization approach proves more effective than O3's sophisticated reasoning. Both agent variants substantially outperform traditional methods, as Bayesian Optimization fails to converge on this challenging task due to the high-dimensional parameter space and complex objective landscape, highlighting the fundamental advantage of reasoning-based approaches over blind optimization in complex battery parameter estimation scenarios.

Table 3: Performance on long-horizon degradation fitting and real-world battery tasks. BO failed to converge and is *excluded* from comparison.

| Method | Degradation | | | | Real Battery | | | |
|--------|------------|--------|--------|--------|--------------|--------|--------|--------|
| | Total MAPE | Q_MAPE | I_MAPE | V_MAPE | Total MAPE | Q_MAPE | I_MAPE | V_MAPE |
| BatterySimAgent-OSS | 1.3674 | 0.5148 | 0.6595 | 0.1931 | 8.7489 | 0.9027 | 6.5803 | 1.2659 |
| BatterySimAgent-O3 (Ours) | 1.7705 | 0.6711 | 0.8501 | 0.2494 | **3.4591** | **0.6020** | **1.8136** | **1.0436** |

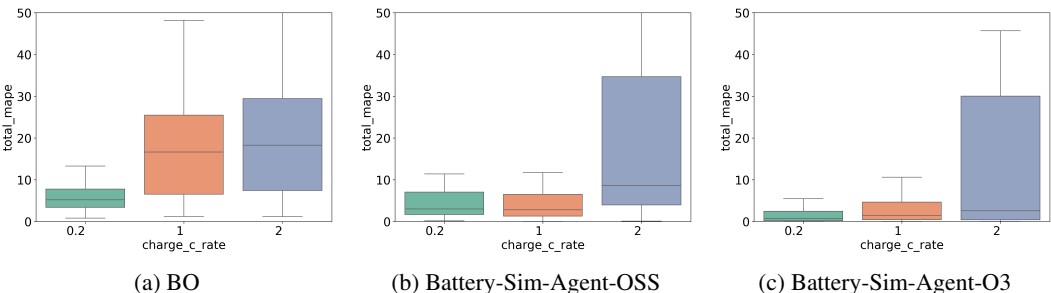

(a) BO      (b) Battery-Sim-Agent-OSS      (c) Battery-Sim-Agent-O3

Figure 3: **Performance across C-rates.** Comparison of different methods across various charge/discharge protocols. Each subplot shows MAPE distribution for different C-rate protocols.

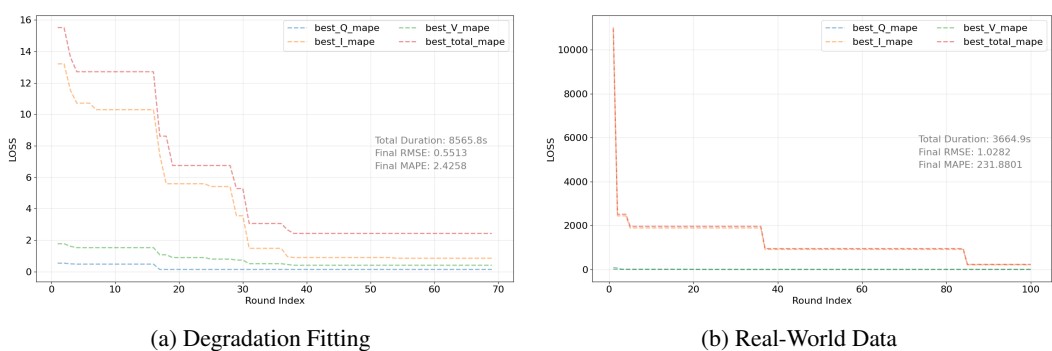

(a) Degradation Fitting      (b) Real-World Data

Figure 4: **Convergence analysis.** Evolution of error metrics over optimization iterations for GPT-O3 on degradation fitting (left) and real-world battery data (right), demonstrating systematic convergence in complex scenarios where traditional methods fail.

**Real-World Validation.** We validate BATTERY-SIM-AGENT on 7 real battery tasks, using data from the CALCEHe et al. (2011); Xing et al. (2013) dataset obtained from public repositories Zhang et al. (2024), demonstrating practical applicability. Figure 4 shows convergence behavior for both degradation fitting and real-world data, revealing robust optimization even with noisy experimental data and unknown ground-truth parameters.

## 6 CONCLUSION AND LIMITATIONS

We introduced BATTERY-SIM-AGENT, a novel framework that reframes the challenging inverse problem of parameterizing battery digital twins as a reasoning task. By deploying an LLM-agent in a closed loop with a high-fidelity simulator, we demonstrated a new paradigm for scientific optimization that mimics human expert workflows. Our comprehensive experiments showed that this reasoning-based approach systematically outperforms traditional black-box optimizers on a diverse suite of simulated benchmarks.

While the results are promising, several aspects merit further investigation. First, in contrast to classical optimization techniques or Bayesian methods with formal guarantees, the LLM-agent's behavior is inherently probabilistic, and its theoretical properties remain an open question. Establishing stronger convergence characterizations is an important direction for future work. Second, the agent's effectiveness naturally depends on the underlying LLM's reasoning capability; exploring lighter-weight or fine-tuned models may further broaden the approach's applicability. Finally, the computational cost of repeated simulation–agent interactions, while manageable, suggests opportunities for efficiency improvements through model reduction or hybrid numerical–agent strategies. These considerations highlight avenues for continued refinement rather than fundamental obstacles. Overall, BATTERY-SIM-AGENT provides a first step toward reasoning-driven autonomous scientific discovery in battery research.

## REPRODUCIBILITY STATEMENT

We have taken several measures to ensure the reproducibility of our results. All experiments were conducted with fixed random seeds, and key experiments were repeated multiple times to verify consistency. Detailed hyperparameter settings are provided in the Appendix. An anonymous repository containing the complete source code, configuration files, and instructions for reproducing all experiments is available at `https://anonymous.4open.science/r/BatteryAgent-BF58/`.

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

## A   USE OF LARGE LANGUAGE MODELS (LLMs)

In preparing this manuscript, we employed a Large Language Model (LLM) as a general-purpose writing assistant. Specifically, the LLM was used to polish the language, improve clarity and flow, and enhance the presentation of the text. All technical content, experimental design, data analysis, and model development were performed independently by the authors. The LLM was not used to generate any novel scientific ideas, experimental results, or interpretations.

## B   BENCHMARK GENERATION DETAILS

### B.1   SINGLE-PARAMETER VARIATIONS (EXTREME MODE)

In this mode, we instantiate the "ground truth" parameter vector $\theta^*$ by applying a large perturbation to a *single* critical parameter from a given base chemistry $\theta_{\text{init}}$. This construction is not the agent's search space, but rather defines the hypothetical battery we want the agent to rediscover through inverse reasoning. Large multiplicative factors are chosen to generate highly non-convex objective landscapes, stress-testing the agent's capability to adapt. The other parameters remain fixed at their base values, preserving physical plausibility.

Table 4 lists the nine parameters and their Perturbation Rules used to generate the Extreme Mode benchmark tasks. Each perturbed parameter set is paired with a fixed base chemistry and protocol, producing a synthetic target battery for evaluation.

Table 4: Parameter perturbation rules for Extreme Mode benchmark. The "base" refers to the unperturbed literature parameter value from $\theta_{\text{init}}$. Factors are multiplicative unless otherwise noted.

| Parameter Name | Perturbation Rule |
| --- | --- |
| Negative particle radius [m] | base $\times\{0.5,\ 2.0\}$ |
| Positive particle radius [m] | base $\times\{0.5,\ 2.0\}$ |
| Negative electrode thickness [m] | base $\times\{0.75,\ 1.5\}$ |
| Positive electrode thickness [m] | base $\times\{0.75,\ 1.5\}$ |
| Negative electrode porosity | base $\pm0.05$ |
| Positive electrode porosity | base $\pm0.05$ |
| Negative electrode Bruggeman coefficient | $\{1.5,\ 2.0,\ 2.5\}$ |
| Positive electrode Bruggeman coefficient | $\{1.3,\ 1.8,\ 2.3\}$ |
| Separator thickness [m] | base $\times\{0.7,\ 1.3\}$ |

### B.2   MULTI-PARAMETER COMBINATIONS (REGULAR MODE)

In this mode, the "ground truth" $\theta^*$ is constructed by applying an *expert-designed combination* of physically plausible perturbations to multiple parameters of a base chemistry. This mimics realistic manufacturing variations or design choices, such as co-varying electrode porosity and thickness to achieve performance trade-offs. The perturbations remain within safe electrochemical limits to avoid simulator instability. As in Extreme Mode, these perturbations are applied only to generate the synthetic target; the agent begins optimization from the unperturbed $\theta_{\text{init}}$.

Table 5 lists the twelve predefined multi-parameter combinations used in Regular Mode. Each combination is paired with a base chemistry and protocol to produce a distinct synthetic target battery.

### B.3   FINAL SELECTION PROCESS

We iterate through all combinations of base parameter sets, C-rates, and perturbation rules from Tables 4 and 5. For each combination, the perturbed parameters define $\theta^*$ and the corresponding simulator output $Y_{\text{obs}}$. The agent starts from the original unperturbed $\theta_{\text{init}}$ and aims to recover $\theta^*$ via iterative reasoning. We apply a two-stage filtering process:

1. **Stability Filter:** Discard parameter combinations that result in simulation failure in PY-BAMM.

Table 5: Predefined multi-parameter combinations for the *regular mode* benchmark.

| ID | Description and Parameter Overrides |
|---|---|
| 1 | **Max-power, manufacturing-plausible:** Neg./Pos. particle radius $\times 0.7$, Neg./Pos. electrode thickness $\times 0.85/0.9$, etc. |
| 2 | **Energy-leaning but realistic:** Neg./Pos. electrode thickness $\times 1.10/1.25$ (maintaining N/P ratio), porosity $-0.02/-0.03$. |
| 3 | **Electrolyte-limited cathode:** Pos. electrode thickness $\times 1.25$, Pos. porosity $-0.05$, Pos. Bruggeman coeff. to 2.0. |
| 4 | **Solid-diffusion-limited (both electrodes):** Neg./Pos. particle radius $\times 1.5$. |
| 5 | **Anode-biased diffusion limit:** Neg. particle radius $\times 1.8$, Neg. electrode thickness $\times 1.15$. |
| 6 | **Cathode-biased diffusion limit:** Pos. particle radius $\times 1.8$, Pos. electrode thickness $\times 1.15$. |
| 7 | **High-$\varepsilon$ / low-tortuosity (ionic-friendly):** Neg./Pos. porosity $+0.06$, Neg./Pos. Bruggeman coeff. to 1.5. |
| 8 | **Low-$\varepsilon$ / high-tortuosity (ionic bottleneck):** Neg./Pos. porosity $-0.06$, Neg./Pos. Bruggeman coeff. to 2.0. |
| 9 | **Asymmetric particles (fast anode / slow cathode):** Neg. radius $\times 0.7$, Pos. radius $\times 1.4$. |
| 10 | **Asymmetric particles (slow anode / fast cathode):** Neg. radius $\times 1.4$, Pos. radius $\times 0.7$. |
| 11 | **Thin separator + thick electrodes:** Separator thickness $\times 0.85$, Neg./Pos. electrode thickness $\times 1.20/1.25$. |
| 12 | **Thick separator + low-$\varepsilon$ (ionic choke):** Separator thickness $\times 1.5$, Neg./Pos. porosity $-0.04$. |

2. **Sensitivity Filter:** Remove cases where the capacity change is less than $1\%$ compared to the baseline.

From the valid cases (233 for Extreme Mode, 373 for Regular Mode), we randomly select 100 tasks per mode to form the final suite of 200 tasks.

### B.4 SIMULATOR STABILITY AND FAILURE MODES

We clarify that the "simulation failures" mentioned in our filtering process refer to non-convergence of the DAE solver (IDAKLU) due to physical infeasibility, rather than numerical precision issues. The DFN model involves coupled non-linear differential-algebraic equations. Certain parameter combinations (e.g., extremely low diffusion coefficients paired with high C-rates) cause state variables such as particle surface concentration to become negative or singular. In these regimes, the electrochemical kinetics (Butler-Volmer equations) become undefined. Since PyBaMM's adaptive solver already minimizes step sizes to machine precision limits to attempt convergence, further manual reduction of resolution or step size does not resolve these fundamental physical singularities. Therefore, we treat these cases as invalid parameter sets.

## C ADDITIONAL EXPERIMENT SETUP

### C.1 LLM-BASED AGENT SETUP

Table 7 summarizes the main hyperparameter settings used for the LLM-based agent, including the number of warm-up steps and the total search budget.

Table 6: Key Hyperparameter Settings of LLM-agent

| Parameter | Value |
|---|---|
| warm-up rounds (warm-up steps $N_w$) | 20 |
| search rounds (budget $T$) | 80 |

### C.2 BAYESIAN OPTIMIZATION EXPERIMENT SETUP

Table 7 lists the key hyperparameters for the Bayesian Optimization experiments, including the optimization platform, random seed, initialization and optimization strategies, surrogate model, and acquisition function.

### C.3 COVARIANCE MATRIX ADAPTATION EVOLUTION STRATEGY EXPERIMENT SETUP

Table 8 presents the main hyperparameters for the CMA-ES experiments, such as random seed, parameter bounds, iteration limits, population size, and various tolerance settings.

Table 7: Key Hyperparameter Settings of Bayesian Optimization

| Parameter | Value |
|---|---|
| Platform | `Meta's Ax (v1.1.0)` |
| Random Seed | `1234` |
| Initialization Strategy | `Sobol sequence` |
| Optimization Strategy | `GPEI` |
| Surrogate Model | `SingleTaskGP (Matern kernel)` |
| Acquisition Function | `LogNEI` |
| Warmup Round | `number of parameters * 2` |

Table 8: Key Hyperparameter Settings of CMA-ES

| Parameter | Value |
|---|---|
| Random Seed | `1234` |
| bounds | `[x0_lower_bounds, x0_upper_bounds]` |
| maxiter | `generations` |
| popsize | `number of parameters + 1` |
| verb_disp | `1` |

# D  ADDITIONAL EXPERIMENTAL RESULTS

## D.1  DETAILED DEGRADATION EXPERIMENT SETUP

For the long-horizon degradation fitting experiments, we select 5 representative parameter sets from our benchmark suite and enable SEI modeling with the "reaction limited" mechanism in PYBAMM. Each simulation runs for 200 cycles to capture capacity fade behavior. The optimization task involves fitting both base electrochemical parameters and SEI degradation parameters (SEI kinetic rate constant, SEI conductivity, etc.) to match the observed capacity degradation curve.

## D.2  REAL-WORLD DATA VALIDATION DETAILS

We apply BATTERY-SIM-AGENT-O3 to 7 real battery datasets from public repositories, including NASA and CALCE battery datasets. These datasets contain charge/discharge cycles from actual lithium-ion batteries under various operating conditions. For each dataset, we use the first few cycles to infer battery parameters and validate against remaining cycles. The convergence analysis demonstrates robust optimization behavior even with noisy experimental data.

## D.3  ADDITIONAL EXPERIMENTS ON WARM-UP STRATEGIES

To validate the design choice of using LLM-generated perturbations during the warm-up phase (Phase 1 of Algorithm 1), we conducted a comparative experiment against a baseline strategy using random fixed perturbations.

**Analysis of Results.**  Figure 5 presents the performance distribution in terms of Total RMSE and Total MAPE. The results demonstrate the superiority of the proposed method:

- **Error Reduction:** The *LLM proposed search* consistently achieves lower median values for both RMSE and MAPE compared to the *Fixed search*. This indicates that the LLM's ability to reason about the initial parameters allows it to identify more promising regions of the search space even during the initialization phase.
- **Stability and Robustness:** As observed in the **Total MAPE** plot (right), the *Fixed search* exhibits a significantly larger spread with upper whiskers extending to high error values (approaching 10.0). In contrast, the *LLM proposed search* maintains a tighter interquartile

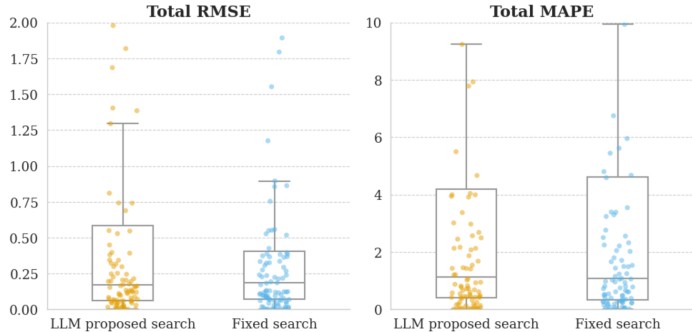

Figure 5: **Comparison of Warm-up Strategies.** The boxplots illustrate the distribution of Total RMSE (left) and Total MAPE (right) achieved by the proposed LLM-driven search (orange) versus a fixed random search strategy (blue). The LLM-driven approach demonstrates lower error metrics and reduced variance.

> range and fewer extreme outliers. This suggests that the LLM-driven warm-up effectively avoids poor parameter configurations that random perturbations might encounter, providing a higher-quality "knowledge memory" for the subsequent main optimization loop.

These findings confirm that the perturbations generated by the LLM are not merely random noise but are purposeful explorations that effectively adapt the model to the current problem instance.

### D.4 ADDITIONAL PERFORMANCE ANALYSIS

**Robustness Analysis.** Our agent's advantage is particularly pronounced in challenging scenarios. In extreme mode, baseline optimizers degrade significantly while our agent remains robust. At higher C-rates (2C), where dynamics are more complex, the performance gap widens further.

**Convergence Behavior.** The convergence analysis reveals that our agent maintains stable optimization behavior even in challenging high-dimensional parameter spaces where traditional optimization methods struggle to converge. This is particularly evident in the degradation fitting task, where BO completely fails to converge.

### D.5 LOSS CURVE ANALYSIS

To provide a more comprehensive evaluation of the BATTERY-SIM-AGENT's optimization process, we present additional convergence curves derived from real-world battery cycling data. While Figure 4 in the main text illustrates a particularly challenging scenario to demonstrate resilience under noise, the results presented here in Figure 6 represent the agent's typical performance characteristics: rapid error reduction and stable convergence to low-error solutions.

**Analysis of Convergence Behaviors.** As shown in Figure 6, the optimization process exhibits a distinct "step-wise" descent pattern, which reflects the LLM's iterative reasoning and parameter decoupling strategy.

- **High-Precision Convergence (Figure 6a):** In this scenario, the agent begins with a high initial total loss (MAPE > 900). We observe sharp reductions in loss around Round 5 and Round 24. This pattern suggests that the agent effectively decouples the parameter space, identifying key physical parameters (such as capacity $Q$ or voltage curve features) sequentially rather than randomly. By Round 44, the agent converges to a highly accurate solution with a final **RMSE of 0.1396** and a **MAPE of 3.22%**, maintaining stability for the remaining rounds.
- **Recovery from Extreme Initialization (Figure 6b):** This case illustrates the agent's robustness against poor initial conditions. The optimization starts with an extremely high

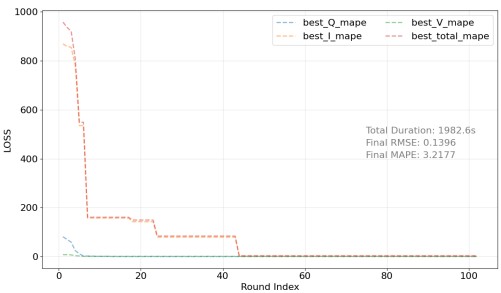 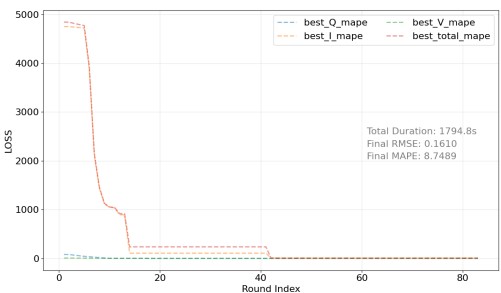

(a) Convergence profile for Sample A. The agent achieves a final MAPE of 3.22%.

(b) Convergence profile for Sample B. The agent achieves a final MAPE of 8.75%.

Figure 6: **Typical Convergence Behaviors on Real-World Data.** The dashed lines represent the Mean Absolute Percentage Error (MAPE) for different parameter groups (Capacity $Q$, Voltage $V$, Loss $L$) and the total loss over optimization rounds. Unlike the stress-test case in the main text, these samples show efficient convergence.

error (MAPE $\approx 4800$). Despite this, the agent quickly identifies the direction of gradient descent, achieving a massive error reduction at Round 8. Subsequent adjustments at Round 15 and Round 42 further refine the parameters. The distinct plateaus between drops indicate the agent exploring local regions before the LLM synthesizes the feedback to propose a new, more effective parameter set. The process concludes with a reasonable physical fit, achieving a final **RMSE of 0.1610** and a **MAPE of 8.75%**.

These additional results confirm that for representative real-world data, the Battery-Sim-Agent is capable of converging significantly faster and achieving much lower final errors than the hard-case example shown in the main text.

## D.6 ABLATION STUDY

### D.6.1 FAILURE CASE STUDY: SENSITIVITY TO POORLY SPECIFIED PRIORS

To investigate the limitations of the BATTERY-SIM-AGENT, we analyze a counter-example (Experiment ID 134) where the agent fails to converge. This case serves as an example of a "wrongly defined prior," where the initial memory provided by the user is incompatible with the target operating conditions.

**Experimental Setup.** The target protocol involves a relatively aggressive 2C CCCV charge and 1C discharge cycle. However, the initial memory provided to the agent is based on the standard ORegan2022 parameter set. As illustrated in Figure 7, this default configuration is numerically unstable under the target high-current protocol.

**Simulation Instability.** Under the default parameters, the PyBaMM solver cannot complete a full charge/discharge cycle. Specifically:

- During the 2C constant current (CC) charge phase, the current remains constant until approximately 600 seconds.

- At this point, the simulation abruptly terminates before entering the constant voltage (CV) phase or the discharge phase.

- This early termination is caused by numerical or physical violations, such as stoichiometry limits, concentration bounds, or Jacobian singularities during the transition.

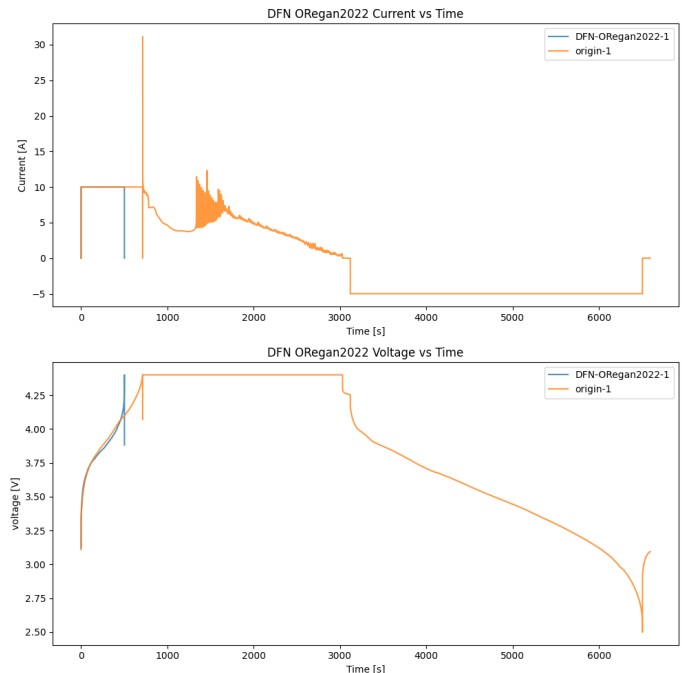

Figure 7: Current–time and voltage–time curves for Experiment ID 134. The orange curve represents the target (ground truth). The blue curve represents the simulation using the default initial memory (prior). The default simulation terminates early ($\approx$ 600s) due to solver failure.

While the modified target parameters (orange curve in Figure 7) allow for a complete cycle, they exhibit strong oscillations in the CV region, indicating that the target landscape itself is highly sensitive to small parameter variations.

**Agent Performance and Analysis.** Figure 8 shows the optimization trajectory over 100 rounds. The Battery-Sim-Agent (based on GPT-OSS) fails to reduce the loss effectively.

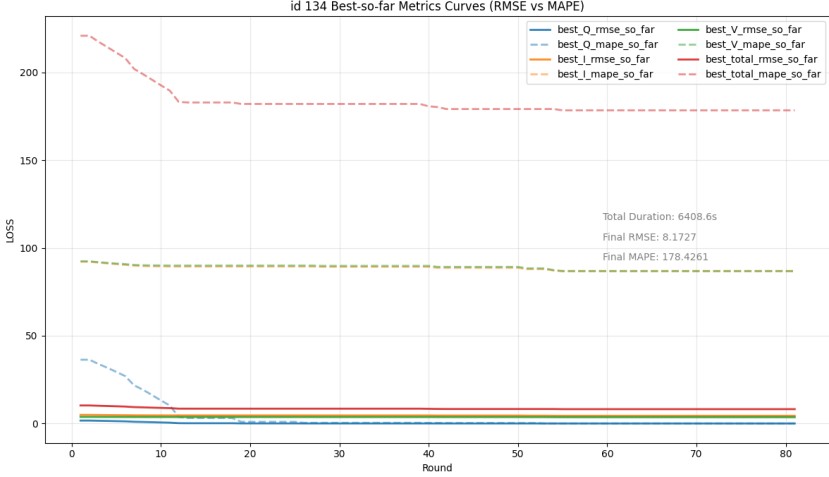

Figure 8: Best-so-far RMSE and MAPE over iterations for Experiment ID 134. The agent fails to converge due to the lack of informative feedback from the environment.

The primary reason for this failure is the lack of informative feedback, leading to a sparse reward problem:

1. **High Crash Rate:** Due to the extreme sensitivity of the configuration, almost every candidate parameter set proposed by the agent triggers a simulation crash. In the initial 20 exploration attempts, only 1 simulation succeeded. Across all subsequent rounds, only 2 additional simulations completed successfully.

2. **Inability to Update Beliefs:** With the majority of evaluations returning solver errors rather than valid loss values, the agent cannot form a meaningful belief over the parameter space.

We observed that this limitation is not unique to our method; GPT-o3 and Bayesian Optimization (BO) baselines also fail to make progress under these conditions. This case study highlights that while LLM-based agents are powerful, they require a prior (initial memory) that is at least physically viable for the target protocol to initiate effective learning.

## E  PROMPT DESIGN OF BATTERY-SIM-AGENT

---

**First-Cycle Calibration Prompt**

**System prompt:**

You are a battery parameter expert with extensive experience and expertise in adjusting battery parameters and are proficient in the PyBaMM simulation tool. You can adjust battery parameters based on the actual battery capacity degradation to ensure that simulation results match actual results.

---

**User prompt:**

**First Round:**

- I want to simulate a real battery using Pybamm, and I plan to adjust the parameters so that the current and voltage curves look consistent.
- The charge and discharge protocol I use is as follows: {{ protocols }}
- I hope you can use your existing knowledge about these parameters and summarize how adjusting these parameters will change the capacity and how the current-voltage curve will change.
- I hope you can adjust the parameters based on your knowledge and these rules, as well as the capacity and curve of the battery under the current parameters, so that the simulated curve is closer to the real one. These are the current parameters.
- You need to adjust these parameters to make the curve of the first circle close. If necessary, you can also change other parameters.
- params = {{ current_params }} And other parameter would follow {{ parameter_set }} parameter set values.
- The upper picture shows the current changing with time curve, and the lower picture shows the voltage changing with time curve. The yellow one is the real battery curve, and the blue one is the curve generated by the {{ model_name }} model with the current parameters. Please first describe the difference between the blue and yellow curves in the figure, and summarize the direction in which the parameters need to be optimized, then adjust the parameters, and return the parameters and values that need to be adjusted.
- {{ cycle_description }}
- We need to ensure that the capacity and the time of different steps (such as constant current charging) are the same between the simulated data and the real data.

---

**User prompt:**

**TEXT KNOWLEDGE:**
And here are some patterns that test by experiment by us:

- For Electrode Width [m], If the value is increased, the corresponding battery capacity will increase, and if the value is decreased, the corresponding battery capacity will decrease.

- For Negative Electrode Active Material Volume Fraction, If the value is increased, the corresponding battery capacity will increase, and if the value is decreased, the corresponding battery capacity will decrease.

- At the same time, decreasing the value will increase the relative proportion of the constant voltage (CV) stage in the charge stage, while the relative proportion of the constant current (CC) stage will decrease.

- Note that the Negative Electrode Active Material Volume Fraction should be larger than the Positive Electrode Active Material Volume Fraction.

- For Positive Electrode Active Material Volume Fraction, If the value is increased and decreased, the corresponding battery capacity will not change significantly.

- At the same time, decreasing the value will increase the relative proportion of the constant voltage (CV) stage in the charge stage, while the relative proportion of the constant current (CC) stage will decrease.

- For Negative Electrode Thickness [m], If the value is increased, the corresponding battery capacity will increase, and if the value is decreased, the corresponding battery capacity will decrease.

- At the same time, decreasing the value will increase the relative proportion of the constant voltage (CV) stage in the charge stage, while the relative proportion of the constant current (CC) stage will decrease significantly. Note that if the value is too large, it will cause errors.

- For Positive Electrode Thickness [m], If the value is increased and decreased, the corresponding battery capacity will not change significantly.

- At the same time, decreasing the value will decrease the relative proportion of the constant voltage (CV) stage in the charge stage, while the relative proportion of the constant current (CC) stage will not change significantly.

- For Maximum Concentration in Negative Electrode [mol.m^-3], If the value is increased and decreased, the corresponding battery capacity will not change significantly.

- At the same time, decreasing the value will increase the relative proportion of the constant voltage (CV) stage in the charge stage, while the relative proportion of the constant current (CC) stage will decrease. Note that the maximum concentration must be greater than the initial concentration.

- For Maximum Concentration in Positive Electrode [mol.m^-3], If the value is increased, the corresponding battery capacity will increase, and if the value is decreased, the corresponding battery capacity will decrease.

- At the same time, increasing the value will decrease the relative proportion of the constant voltage (CV) stage in the charge stage, while the relative proportion of the constant current (CC) stage will decrease.

- Note that the maximum concentration must be greater than the initial concentration, and slight adjustments may cause errors.

- For Initial Concentration in Negative Electrode [mol.m^-3] If the value is increased, the corresponding battery capacity will increase, and if the value is decreased, the corresponding battery capacity will decrease.

- At the same time, decreasing the value will decrease the relative proportion of the constant voltage (CV) stage in the charge stage, while the relative proportion of the constant current (CC) stage will decrease.

- For Initial Concentration in Positive Electrode [mol.m^-3], If the value is increased, the corresponding battery capacity will decrease, and if the value is decreased, the corresponding battery capacity will increase.

- At the same time, decreasing the value will decrease the relative proportion of the constant voltage (CV) stage in the charge stage, while the relative proportion of the constant current (CC) stage will decrease.

The params should just in {{ search_keys }}, I hope you can summarize the above results and suggest the next 1 updated params group with new values as dict (without name) in JSON format.

**SEARCH KNOWLEDGE:**

I want to explore and gather the knowledge from first 20 groups results. You can only modify some parameters in this list {{ search_keys }}. Give me a series of parameter adjustment (about 20 groups) in JSON format for me to execute using pybamm first. Please do not add any invalid comments for JSON.

**OTHER ROUNDPROMPT:**

Here are the results:

{{ cycle_description }}

The params should just in {{ search_keys }}, I hope you can summarize the above results and suggest the next 1 updated params group with new values as dict (without name) in JSON format.

---

Long-Horizon Degradation Fitting Prompt

**System prompt:**

You are a battery parameter expert with extensive experience and expertise in adjusting battery parameters and are proficient in the PyBaMM simulation tool. You can adjust battery parameters based on the actual battery capacity degradation to ensure that simulation results match actual results.

---

**User prompt:**

**First Round:**

- I want to simulate a real battery degradation using Pybamm, and I plan to adjust the SEI parameters so that the current and voltage curves of every cycle look consistent.

- The initial settings are the same, so the first cycle of real and simulated data are the same. From cycle 2, we want to adjust SEI params to keep real and simulated data look same. I will provide the corresponding cycle number {{ cycle_idxs }} and the corresponding real and simulated information.

- The charge and discharge protocol I use is as follows: {{ protocols }}

- I hope you can use your existing knowledge about these parameters {{ search_keys }} and summarize how adjusting these parameters will change the degradation capacity and how the current-voltage curve will change.

- # I hope you can adjust the parameters based on your knowledge and these rules, as well as the capacity and curve of the battery under the current parameters, so that the simulated curve is closer to the real one.

- You need to adjust these parameters to make the curve of the {{ cycle_idxs }} close.

- curent params = {{ current_params }} and other parameter would follow {{ parameter_set }} parameter set values.

- {{ cycle_description }}

- We need to ensure that the capacity and the time of different steps (such as constant current charging) are the same between the simulated data and the real data of each cycle.

---

**User prompt:**

**TEXT KNOWLEDGE:**
And here are some patterns that test by experiment by us:

- Higher solvent concentration (bulk_solvent_concentration_mol_m-3) accelerates side reactions like the SEI, leading to greater degradation.

- A higher lithium-to-SEI molar ratio (ratio_of_lithium_moles_to_SEI_moles) increases the active lithium consumption efficiency and accelerates capacity degradation.

- Increasing the initial EC concentration in the electrolyte (EC_initial_concentration_in_electrolyte_mol_m-3) generally results in larger initial capacity and impedance decay, with a downward-convex curve.

- A higher SEI solvent diffusivity (SEI_solvent_diffusivity_m2_s-1) increases the degradation rate and magnitude.

- A higher EC diffusivity (EC_diffusivity_m2_s-1) accelerates the degradation rate and results in a downward-convex curve.

- A higher initial SEI thickness (initial_SEI_thickness_m) slows the degradation rate and minimizes the degradation. The larger the SEI partial molar volume (SEI_partial_molar_volume_m3_mol-1), the slower and larger the degradation.

    The params should just in {{ search_keys }}, I hope you can summarize the above results and suggest the next 1 updated params group with new values as dict (without name) in JSON format.

The params should just in {{ search_keys }}, I hope you can summarize the above results and suggest the next 1 updated params group with new values as dict (without name) in JSON format.

**SEARCH KNOWLEDGE:**
I want to explore and gather the knowledge from first 10 groups results. You can only modify some parameters in this list {{ search_keys }}. Give me a series of parameter adjustment (about 10 groups) in JSON format for me to execute using pybamm first. Please do not add any invalid comments for JSON.

**OTHER ROUNDPROMPT:**
Here are the results:
{{ cycle_description }}
The params should just in {{ search_keys }}, I hope you can summarize the above results and suggest the next 1 updated params group with new values as dict (without name) in JSON format.

