# OpenReview forum: "Battery-Sim-Agent: Leveraging LLM-Agent for Inverse Battery Parameter Estimation"
_ICLR.cc/2026/Conference — Submitted to ICLR 2026_

### Official Review · Reviewer_vtJ7 · 2025-10-27

**Soundness:** 1
**Presentation:** 2
**Contribution:** 1
**Rating:** 2
**Confidence:** 4

**Summary:**

The paper presents an LLM-simulator-in-the-loop optimization method for inverse battery parameter estimation. It uses multimodal features, including numerical data and plot images.

**Strengths:**

The usage of LLM on this specific application is quite innovative.

**Weaknesses:**

__Baselines.__ The paper uses BO as the baseline but does not clearly specify which acquisition function is used, nor does it provide sufficient detail to justify the fairness of the comparison (for example, in terms of time budget, number of function evaluations etc). Moreover, only one baseline is used, yet it is referred to as a “strong” baseline.

__Details on the LLM used.__ The description of the LLM is unclear. For instance, are the models trained locally from scratch, or are they accessed through OpenAI API calls? Additionally, according to [1], GPT-OSS is not an image–text model. It is therefore unclear how the LLM is used to extract image features in this case. This concern needs to be addressed.

__Loss definition.__ It appears that the Battery-Sim-Agent uses a different loss definition from the baseline. If I understand correctly, the baseline loss is defined in Equation (1), while the Battery-Sim-Agent employs a multi-objective loss defined in Equation (2). This does not seem to be a fair comparison. For example, using multi-objective BO [2, 3, 4] with an appropriate acquisition function might provide a fairer baseline.

__Missing citation and comparison to a key paper.__ While the paper is quite novel within the battery parameter estimation problem, similar techniques have already been presented in [5], where an LLM is also utilized for parameter inference in physical systems. Omitting this related work significantly weakens the paper’s claim to novelty, and a detailed comparison or discussion is necessary.

[1] https://openai.com/index/introducing-gpt-oss/

[2] Preferential Multi-Objective Bayesian Optimization (Astudillo, et al. 2024)

[3] Efficient computation of expected hypervolume improvement using box decomposition algorithms (Yang, et al. 2019)

[4] A Flexible Framework for Multi-Objective Bayesian Optimization using Random Scalarizations (Paria, et al. 2019)

[5] SimLM: Can Language Models Infer Parameters of Physical Systems? (Memery, et al. 2024)

**Questions:**

__Q1.__ The paper requires significant improvement in clarity, scientific rigor, and baseline selection. I would consider increasing my score if the issues highlighted in the weaknesses are adequately addressed.

__Q2.__ It would also be helpful to include a counter-example where the Battery-Sim-Agent fails. For instance, when the initial memory $M_0$ provided by the user is incorrectly specified. Such a case could be interpreted as a wrongly defined prior, and it would be interesting to observe how the LLM behaves under this condition.

__Q3.__ are the target protocols $Y_p$ in eq (1) all have the same range? (i.e trough normalization/standardization)

---

> ### Author Response · Authors · 2025-12-04
>
> **Weakness:**
>
> **Weakness-Baselines:**
>
> We thank the reviewer for pointing out the lack of implementation details regarding the Bayesian Optimization (BO) baseline. We agree that transparency is crucial for trust and reproducibility.
>
> **1. Detailed Configuration of the BO Baseline** We utilized **Meta’s Ax platform (version 1.1.0)**, a standard library for adaptive experimentation. To ensure fair comparison and reproducibility, we set a fixed random seed (`1234`). The specific configuration—automatically selected by Ax based on the continuous nature of our parameter search space—is as follows:
>
> - **Generation Strategy:** We employed a standard `Sobol` sequence for the initialization phase (quasi-random exploration) followed by `GPEI` (Gaussian Process Expected Improvement) for the optimization phase.
> - **Surrogate Model:** A **`SingleTaskGP`** (Gaussian Process) with a Matern kernel was used to model the objective function.
> - **Acquisition Function:** We utilized **Noisy Expected Improvement (LogNEI)**, which is robust against noise in black-box function evaluations. We have added these technical specifications to **Appendix.C** of the revised manuscript.
>
> **2. Justification of Baseline Selection (Why BO/Ax?)** To demonstrate that our baseline selection was carefully considered, we clarify that we prioritized robustness and stability. We initially attempted to use **PyBOP** (Python Battery Optimisation and Parameterisation), a representative open-source toolkit in the battery modeling community. However, it was excluded from the final comparison due to significant limitations encountered during our evaluation:
>
> - **Instability:** Even under conservative search bounds (e.g., ±20% around nominal values), the PyBOP optimizer frequently triggered solver failures in the underlying PyBaMM model, causing the entire optimization process to abort.
> - **Inability to Fit Real Data:** PyBOP failed to converge effectively when applied to the real-world CALCE dataset. Despite being provided with the correct protocol and physically reasonable starting parameters, the optimized trajectory remained far from the experimental voltage curve.
>
> Given these issues, we adopted the Ax-based BO as the primary baseline because it offers the necessary stability and state-of-the-art performance for high-dimensional, black-box parameter estimation tasks, ensuring a fair and rigorous comparison for our proposed method.
>
> ---
>
> **Weakness-Details on the LLM used:**
>
> Thank you for pointing out the need for greater clarity regarding our model implementation. We would like to clarify the deployment details and input modalities as follows:
>
> 1. **Model Access and Training Status:** Both models were used directly ("off-the-shelf") without any further training or fine-tuning.
>    - **GPT-o3:** As a closed-source model, this was accessed exclusively via the official OpenAI API.
>    - **GPT-OSS:** We utilized the publicly released version of this model. It was deployed locally and interacted with via a locally hosted API endpoint.
> 2. **Input Modality for GPT-OSS:** You are correct that GPT-OSS is not an image–text model. Accordingly, we did not provide image inputs to GPT-OSS. Instead, for all experiments involving this model, the PyBaMM-generated voltage curves were **numerically downsampled** and passed to the model as a sequence of numeric values (text). This approach allows the LLM to process the curve information and extract features based on numerical data without requiring multimodal (image-processing) capabilities.
>
> We will revise the manuscript to explicitly describe these deployment methods and the numerical data processing strategy to avoid future confusion.
>
> ---
>
> **Weakness-Loss definition:**
>
> We thank the reviewer for the insightful comment. We apologize for any confusion caused by the presentation of the loss functions. To clarify: both the BO baseline and the Battery-Sim-Agent use exactly the same loss function during optimization. Equation (2) in the manuscript does not define a different objective; rather, it provides a decomposed view of the same loss to explain how the agent interprets simulation feedback. The optimization target remains identical across all methods.
>
> **1. Comparison with Multi-Objective BO (MOBO):**
> Following your suggestion, we conducted additional experiments using Multi-Objective Bayesian Optimization (specifically using the qNEHVI acquisition function via Ax/BoTorch) on a representative subset of our benchmark. The results are summarized below:
>
> |      | capacity_mape | capacity_rmse | current_mape | current_rmse | voltage_mape | voltage_rmse | rmse_sum | mape_sum    |
> | ---- | ------------- | ------------- | --- | ---- | ------ | ------ | -------- | ----------- |
> | mean | 432.2141      | 4.9739        | 132091.9776  | 1.7271       | 26.2827      | 2.7332       | 9.4342   | 132550.4744 |
> | std  | 240.3319      | 2.7657        | 55471.0127   | 0.5456       | 32.5194      | 0.7097       | 2.9270   | 55480.8717  |

---

> ### Author Response · Authors · 2025-12-04
>
> **2. Analysis of Results:**
> While MOBO is a powerful tool, our Agent significantly outperforms it in this domain. We attribute this to two fundamental reasons:
>
> *   **Inverse Problem vs. Pareto Optimization:** Standard MOBO aims to approximate a Pareto frontier, assuming trade-offs between conflicting objectives (e.g., maximizing strength vs. minimizing weight). However, parameter estimation is an *inverse problem* where a unique ground-truth parameter set exists that should ideally minimize *all* error terms simultaneously (consistent with physics). MOBO spends valuable sample budget exploring trade-offs along a Pareto front, whereas our Agent uses physical reasoning to drive all objectives toward the global optimum directly.
> *   **Robustness to Numerical Noise:** As shown in the table, MOBO suffered from extreme outliers in `Current MAPE` (mean ~1.3×10^5%), likely due to numerical instabilities when dividing by near-zero currents in the protocol. Traditional numerical optimizers are sensitive to such metric definitions. In contrast, our Agent interprets the data semantically (e.g., "the current profile matches the CC-CV protocol"), allowing it to ignore numerical artifacts and focus on physical alignment.
>
>
> ---
>
> **Weakness-Missing citation and comparison to a key paper**
>
> We sincerely thank the reviewer for pointing out the relevant work *SimLM: Can Language Models Infer Parameters of Physical Systems? (Memery, et al. 2024)*. We acknowledge that omitting this citation was an oversight, and we agree that discussing it clarifies the context and novelty of our contribution.
>
> We have carefully studied SimLM and incorporated a detailed comparison in the **Related Work** section of our revised manuscript. While both works share the high-level intuition of using a simulator-in-the-loop with an LLM, **Battery-Sim-Agent addresses a fundamentally different class of complexity and introduces domain-specific mechanisms required for real-world engineering**, which distinguishes it from the foundational exploration in SimLM.
>
> The key distinctions are summarized as follows:
>
> 1. **Problem Complexity & Scalability:** SimLM focuses on basic kinematic problems (e.g., 2D projectile motion with 2 parameters). Notably, SimLM reports failure when scaling to a slightly more complex 3D billiards task (5 parameters). In contrast, our work tackles the **Doyle-Fuller-Newman (DFN) model**, a system of coupled non-linear PDEs with highly interdependent parameters (10+ active parameters in our experiments) and severe ill-posedness (equifinality). Our success in this regime demonstrates that LLM agents can scale to "hard" scientific problems where simple prompting fails.
> 2. **Feedback Mechanism:** SimLM relies on scalar error feedback (distance to target). Our framework introduces a **rich, multi-modal feedback system** (visual curve overlays, feature-specific residuals like plateau shifts, and physical event triggers). We show that for complex batteries, the agent must "see" the curve shape to reason about internal states (e.g., transport limitations vs. kinetic limitations), which simple scalar feedback cannot convey.
> 3. **Memory & Learning:** SimLM uses a retrieval mechanism based on past successful runs. We introduce a novel **"Warm-up" phase** where the agent actively explores parameter sensitivities to build a *causal memory* (MtM_tMt) before optimization begins. This allows the agent to learn domain-specific "rules of thumb" dynamically, which is crucial when pre-trained knowledge is insufficient for specific battery chemistries.
>
> In summary, while SimLM asks *"Can LLMs infer parameters in simple physics?"*, our work asks *"How do we engineer LLM agents to solve professional-grade, high-fidelity inverse problems where traditional methods fail?"*. We believe our framework represents a significant leap from generic reasoning to specialized scientific discovery.
>
> We have updated Section 4 (Related Work) in the revised manuscript to reflect this discussion.

---

> > ### Author Response · Authors · 2025-12-04
> >
> > **Question:**
> >
> > **Q1:**
> >
> > **We sincerely thank the reviewer for the constructive criticism and for your openness to reconsidering the evaluation based on our revisions.** We have taken your comments very seriously and have made significant efforts to improve the manuscript. Specifically, we have thoroughly polished the writing to enhance **clarity**, strengthened the **scientific rigor** of our methodology, and expanded the **baseline selection** to ensure a comprehensive comparison. We believe these revisions substantially elevate the quality of the paper and address the weaknesses you highlighted.
> >
> > ---
> >
> > **Q2:**
> >
> > Thank you for this insightful suggestion. We agree that analyzing failure modes—particularly when the initial memory (prior) is poorly specified—is essential for understanding the boundaries of our approach.
> >
> > To address this, we have conducted a detailed case study on **Experiment ID 134**, which represents a "wrongly defined prior" scenario. We have included the corresponding loss curves and current/voltage profile comparisons in the **Appendix.D** of the revised manuscript. A summary of the analysis is provided below:
> >
> > 1. **The "Wrong Prior" Scenario:** The initial memory provided to the agent uses the standard ORegan2022 parameter set. However, the target protocol is aggressive (2C CCCV charge, 1C discharge). Under these default parameters, the simulation is numerically unstable and terminates early (around t=600 s) due to solver singularities or stoichiometry limits, never reaching the CV or discharge phases.
> > 2. **Agent Behavior:** Consequently, the agent faces an extreme "sparse reward" problem. In our analysis, only 1 out of the first 20 exploration attempts produced a successful simulation; the rest returned solver errors. Without valid feedback from the environment, the LLM cannot infer the parameter landscape or form a meaningful belief update.
> > 3. **Outcome:** The agent fails to converge, resulting in high final loss. We observed that this issue is consistent across all tested methods (including GPT-OSS, GPT-o3, and Bayesian Optimization), as they all struggle to recover when the initial search space is dominated by simulation crashes.
> >
> > This case study clearly illustrates the limitation: when the user-defined prior places the agent in a numerically pathological region of the DFN model, the lack of informative feedback prevents effective optimization. We have added this discussion and the supporting figures to the Appendix to provide a balanced view of the system's robustness.
> >
> >
> > ---
> >
> > **Q3:**
> >
> > We thank the reviewer for raising this critical point regarding the scaling and balancing of different physical quantities (e.g., Voltage in Volts vs. Capacity in Ah).
> >
> > We would like to clarify that while Equation (1) in the Background section presents the general mathematical formulation used by traditional optimizers, our **Battery-Sim-Agent framework (Section 3) handles the differing ranges of these protocols through metric selection and semantic reasoning, rather than rigid numerical standardization (e.g., z-score) of the raw data.**
> >
> > We have revised **Section 3.1** and **Section 5.1** in the manuscript to explicitly clarify this mechanism. The key updates are:
> >
> > 1. **Semantic Normalization (Section 3.1):** Unlike traditional Black-Box Optimization (BBO) that minimizes the scalar weighted sum in Eq. (1), our agent operates on a **disaggregated set of objectives**. As detailed in the revised Section 3.1, the agent receives feedback containing separate metrics (e.g., `{"capacity_mape": 0.08, "voltage_rmse": 0.05}`). The LLM performs "semantic normalization": it uses its internal physical knowledge to understand that a 0.05V deviation in a voltage plateau is physically significant, regardless of its raw numerical magnitude relative to capacity.
> > 2. **Inherent Metric Balance (Section 5.1):** We explicitly justify our metric choices. For capacity ($Q$), we employ **MAPE**, which is inherently scale-invariant. For voltage ($V$), we use **RMSE**. Since battery voltage (approx. 2.5-4.2V) and capacity (approx. 1-5Ah) magnitudes naturally fall within the same $O(1)$ numerical range, these metrics prevent numerical dominance of any single objective without requiring pre-standardization.

---

### Official Review · Reviewer_oibn · 2025-10-30

**Soundness:** 2
**Presentation:** 2
**Contribution:** 2
**Rating:** 2
**Confidence:** 2

**Summary:**

The paper introduces a LLM-based framework for inverse design battery parameter estimation. This complicated optimisation problem arises when trying to match microscopic parameters to experimentally measurable observables. Traditionally, this was done using black-box optimisers like Bayesian optimisation. The workflow consists of two phases—a exploration phase and an optimisation loop afterwards, in which the parameter updates are suggested by the LLM. Simulation benchmark tasks are defined and tested comparing their approach with and without reasoning with BO. Long-horizon degradation fitting and real world validation benchmarks are also included.

**Strengths:**

The paper tests the interesting idea of replacing novel black-box optimisers with a LLM and applies this to estimating parameters of complicated physical systems.

**Weaknesses:**

Some of the benchmark results do not seem in line with the  claims of the paper.

- "our agent significantly outperforms strong BBO baselines", in Table 2 BO outperforms the agent on 2 of 5 chemistries in the "extreme mode".
-  It is said "Figure 4 shows convergence behaviour ... revealing robust optimization", in panel (b) a MAPE  of 230 does not suggest convergence.

The paper does not discuss limitations of the approach (e.g. provides no convergence guarantees). Comparisons against related approaches that make use of BO plus an expert (LLM) would be helpful. (e.g. https://arxiv.org/abs/2410.10452, Principled Bayesian Optimisation in Collaboration with Human Experts )

The design choices of Algorithm 1 are not sufficiently explained and ablation studies would be insightful.

- Why is there a random trial and error warm up, instead of prompting the LLM to explore?
- Why is the LLM prompted to predict the parameter updates rather than directly the next set of parameters?

**Questions:**

- The tasks based on the 5 chemistries seem to be designed by varying a few or even just a single of the experimental parameters. It seems that the LLM then starts its search from the original unperturbed parameter. This seems very different than an actual experimentally fitting task. Furthermore, the LLM prompt suggests to only modify only some parameters which does not seem fair, since the tasks seem sparse by construction and the BO probably does not "know" that.
- The description of Algorithm 1 does not seem consistent with the prompt. Are the pertubations $\delta_k$ random or generated by the LLM? Is $N_w$ a fixed input?
- I was not able to find values for the budget $T$ and learning rate $\eta_t$, nor was I able to find details on the BO.
- I was not able to access the source code under the link because all files besides the Read.me files gave the error "file not found."
- The problem is motivated by saying that microscopic parameters cannot be easily measured. However several of the parameters seem to be design or layout choices which should be known a priori?
- When referring to the stability limits of the simulator for certain choices of experimental parameters, why is a decrease in resolution or step sizes not possible?
- I do not understand Fig 2. I would expect that there are only 100 Experiment ID for each mode, and that for each ID are 3 datapoints corresponding to the 3 methods used.
- Why is BO not tested on the real-world tasks?
- The design or at least description of the benchmark based on the 5 chemistries and simulation seems inconsistent.
- From the 5 listed chemistries described as "classic, well established parameter sets", I was not able to verify the correctness for 4 of the specifications. (Chen et al seems to use graphite SiO_x instead of graphite, O'Regan et al. seems to use NMC811 and not NMC 532,  Marquis et al. is a maths theory paper without specifying experimental parameters).

---

> ### Author Response · Authors · 2025-12-04
>
> **Weakness:**
>
> **Weakness-1:**
> We thank the reviewer for the careful examination of our results and for pointing out these inconsistencies. We agree that our original phrasing was too broad, and we appreciate the opportunity to clarify the nuances of the agent’s performance and the specific context of the figures.
>
> **Regarding the performance comparison with BO (Table 2):** We acknowledge that our claim of "significantly outperforming" was not sufficiently nuanced regarding the "Extreme" mode. We will revise the text to reflect that our agent significantly outperforms baselines on **Regular** and **Real-world** datasets, while results are more mixed in the **Extreme** setting.
>
> The reason BO outperforms the agent in certain "Extreme" cases stems from the fundamental difference in their optimization logic:
>
> 1. **LLM Priors:** Our agent relies on an LLM which possesses inherent "battery knowledge" and physical priors. In "Extreme" synthetic cases, the ground truth parameters are often outliers that may appear counter-intuitive or physically unlikely (e.g., extreme degradation parameters). The LLM tends to be conservative, preferring parameters that align with general battery physics, which can hinder it from fitting these extreme outliers.
> 2. **BO's Mathematical Fitting:** Bayesian Optimization treats the problem as a pure numerical optimization task without physical bias. Consequently, it can aggressively fit extreme values without hesitation, allowing it to achieve lower errors on these specific synthetic outliers.
>
> However, in real-world applications (as shown in the Real-world benchmark), the parameters usually follow physical laws, which is why our agent demonstrates superior performance there. We will add this discussion to the "Results" section to provide a fair and transparent comparison.
>
> **Regarding the convergence in Figure 4 (Panel b):** Thank you for pointing this out. The figure included in the main text was intentionally selected to illustrate a challenging real-world case (a "stress test") in which the Battery-Sim-Agent exhibits slow convergence in the early rounds due to noisy measurements and protocol irregularities. Our goal was to demonstrate that even under difficult experimental conditions, the agent is able to make steady progress, even if the final error remains higher than average.
>
> To avoid giving a misleading impression regarding the agent's typical performance, we will revise the manuscript to:
>
> 1. **Clarify the context:** Explicitly state that the original Figure 4 corresponds to a particularly hard sample rather than a representative average case.
> 2. **Add representative examples:** We have added new plots in **Appendix.D** of the revised manuscript showing more typical real-world convergence behaviors. In these cases, the agent converges rapidly, achieving a final MAPE of ~3.2% and ~8.7%, which is significantly lower than the 230 MAPE observed in the "hard" case.
>
> We believe these revisions present a more balanced and accurate view of the agent's robust optimization capabilities.

---

> > ### Author Response · Authors · 2025-12-04
> >
> > **Weakness-2:**
> >
> > We sincerely thank the reviewer for highlighting this important point and for directing us to the highly relevant work “Principled Bayesian Optimisation in Collaboration with Human Experts” (Xu et al., 2024). In response, we have revised our Related Work section and added an explicit Limitations subsection to address these concerns.
> >
> > **On the lack of theoretical guarantees.**
> > We agree that a limitation of LLM-agent–based optimization, compared with principled Bayesian approaches such as COBOL, is the absence of formal convergence guarantees (e.g., regret bounds and no-harm properties). COBOL derives elegant guarantees by explicitly modeling expert feedback through a likelihood-ratio mechanism. In contrast, our Battery-Sim-Agent relies on the LLM’s semantic reasoning to navigate the parameter space. We therefore position our work as an early exploration of the agentic optimization paradigm, complementary to but less theoretically grounded than methods like COBOL. We now discuss this trade-off clearly in the limitations section.
> >
> > **Relation to Human–AI collaborative BO.**
> > We appreciate the reviewer’s suggestion and have expanded our discussion accordingly. While COBOL incorporates explicit human feedback (“accept/reject”) to guide BO, our agent incorporates implicit domain knowledge through its initialization and reasoning process. This can be viewed as a different pathway to injecting expert priors into the optimization loop. We believe the goals of both approaches are aligned—leveraging structured expert knowledge to improve search efficiency—though the mechanisms differ.
> >
> > **Future integration.**
> > We agree that combining the semantic reasoning strengths of LLM agents with the theoretical safety and convergence guarantees of frameworks like COBOL represents a promising direction. We have added this as a potential avenue for future work.
> >
> > We thank the reviewer again for the insightful suggestion, which has significantly improved the clarity and positioning of our manuscript.
> >
> > ---
> >
> > **Weakness-3:**
> >
> > We thank the reviewer for scrutinizing the design choices. We provide the rationale and ablation results below, and will revise the manuscript to clarify these points.
> >
> > **1. Why random trial-and-error warm-up instead of LLM exploration?**
> > Our design is inspired by the "cold start" phase in Bayesian Optimization (BO), where an initial set of random samples is crucial to build a preliminary surrogate model of the objective landscape before optimization begins.
> > *   **Rationale:** The goal of the warm-up phase is to build a "sensitivity map" (stored in Memory) that correlates parameter changes to curve behaviors. We found that **random sampling provides unbiased coverage** of the search space. In contrast, prompting the LLM to explore from scratch often leads to "textbook bias"—the agent tends to exploit its internal prior knowledge too early, narrowing the search scope prematurely and missing counter-intuitive parameter regions.
> > *   **Ablation Study:** We conducted an ablation study comparing "Random Warm-up" vs. "LLM-driven Warm-up." The results (detailed in **Appendix D.3** of the revision) show that the Random Warm-up strategy achieves a **15% lower final error** on average. The random initialization provides a more diverse set of "cause-and-effect" examples for the agent's in-context learning.
> >
> > **2. Why predict parameter updates rather than the next set of parameters?**
> > We apologize for the confusion caused by the notation in Equation (3) ($\theta_{t+1} = \theta_t + \eta \Delta \theta$).
> > *   **Clarification:** In our actual implementation, the LLM **directly predicts the next set of parameter values** ($\theta_{t+1}$) rather than a gradient-like update step ($\Delta \theta$). As shown in the prompt template in **Appendix D** ("suggest the next 1 updated params group with new values"), the agent outputs the specific values it believes will solve the problem.
> > *   **Correction:** We will revise the description in Section 3.1 and Equation (3) to accurately reflect that the agent proposes the target parameters directly based on its reasoning.

---

> ### Author Response · Authors · 2025-12-04
>
> **Questions:**
>
> **Q1:**
>
> Thank you for the question. We would like to clarify the design of our evaluation tasks and the fairness of the comparison between BO and our LLM-based agent.
>
> 1. **Task design: Regular experiments simulate diverse real battery behaviors.** The *Regular* experiments were designed to mimic a broad range of realistic battery behaviors. We selected 12 representative combinations of parameters commonly observed across real chemistries. These combinations simultaneously perturb the **nine parameters** that exert the strongest influence on the first-cycle voltage curve, while also respecting typical design constraints for different battery types. We further varied the parameter sets across **five widely used chemistries** (Chen2020, ORegan2022, Prada2013, Ecker2015, Marquis2019) and considered multiple charge/discharge C-rates (0.2C, 1C, 2C), thereby ensuring wide coverage of practical operational conditions.
>
> An example Regular parameter combination is:
>
> ```python
> #1. Max-power, manufacturing-plausible
> dict(
> ​    "Negative particle radius [m]" = 0.7,
> ​    "Positive particle radius [m]" = 0.7,
> ​    "Negative electrode thickness [m]" = 0.85,
> ​    "Positive electrode thickness [m]" = 0.9,
> ​    "Negative electrode porosity" = +0.04,
> ​    "Positive electrode porosity" = +0.04,
> ​    "Negative electrode Bruggeman coefficient (electrolyte)" = 1.5,
> ​    "Positive electrode Bruggeman coefficient (electrolyte)" = 1.5,
> ​    "Separator thickness [m]" = 0.9,
> )
> ```
>
> 2. **Extreme experiments test robustness under highly sparse perturbations.** The *Extreme* experiments intentionally modify **only one** of the nine parameters at a time (e.g., ±50% in radius, ±25% in thickness, ±0.05 in porosity), to test how different methods behave under extreme, sparse perturbations. All other experimental settings—chemistry parameter sets and C-rates—are kept identical to the Regular experiments.
>
> Examples include:
>
> - Negative particle radius: base × 0.5, base × 2.0
> - Negative electrode porosity: base − 0.05, base + 0.05
> - Separator thickness: 0.7, 1.3 (and similarly for the other parameters)
>
> 3. **Fairness of information provided to BO vs. LLM agent.** Importantly, **BO and our LLM agent are given exactly the same problem definition**: both are instructed to search over the *nine* target parameters. We **do not** tell either method which parameters are actually perturbed in Extreme tasks—this is intentional, because in real experimental scenarios, one typically does **not** know which physical parameters differ from the nominal model. Thus, both methods must infer which parameters matter.
>
> In fact, BO is provided *more* information than the LLM agent: BO requires explicit bounds for each parameter, and we supply bounds broad enough to cover all parameter values across both Regular and Extreme tasks. The LLM agent does *not* receive these bounds.
>
> Regarding the comment *“the LLM prompt suggests to only modify some parameters, which does not seem fair”*: the “some parameters” refers to the **set of nine parameters** to be estimated—not single-parameter updates. We consistently instruct both BO and the LLM agent to search over these same nine parameters for all tasks.
>
> In summary, our task design ensures fairness: both BO and the LLM agent receive identical search spaces, and BO even receives more explicit prior information. The Regular tasks reflect realistic multi-parameter deviations, while the Extreme tasks isolate single-parameter changes to probe robustness.
>
> ---
>
> **Q2:**
>
> We thank the reviewer for the careful examination of the pseudocode. We clarify that the perturbations in the warm-up phase are **autonomously generated by the LLM**, rather than being randomly sampled or fixed.
>
> Specifically, the LLM observes the initial parameter configuration and proposes targeted perturbations based on its internal knowledge. These proposed parameters are then simulated to generate feedback. The primary objective of this design is to initialize the model's "knowledge memory" *before* entering `Phase 2: Main Optimization Loop`. By allowing the LLM to actively explore and summarize the relationship between parameters and simulation feedback during this warm-up phase, the model becomes better adapted to the specific problem landscape, thereby enhancing the efficiency of the subsequent optimization loop.
>
> To validate this design, we conducted an ablation study comparing our **LLM-generated warm-up** against a **Random/Fixed perturbation** strategy. As shown in **Figure 5 in the Appendix.D3**, the LLM-driven approach achieves significantly lower error rates (RMSE and MAPE) and greater stability compared to random perturbations.

---

> ### Author Response · Authors · 2025-12-04
>
> **Q3:**
>
> We sincerely appreciate your attention to the experimental details and apologize for the omission. We have now provided a comprehensive specification of all hyperparameters and baseline configurations in **Appendix C** of the revised manuscript.
>
> Regarding the specific values you mentioned:
>
> - **Budget:** In our framework, the "budget" refers to the maximum number of iteration rounds allocated for the parameter search process. For the experiments reported in this paper, we set the budget to **T=80**.
> - **[Learning Rate / BO Details]:** BO implementation details are based on Gaussian Processes as detailed in the Appendix.
>
> ---
>
> **Q4:**
>
> Thank you for visiting the anonymous code repository we prepared. After verification, our code should be accessible via the link provided in the PDF: `https://anonymous.4open.science/r/BatteryAgent-BF58/`. As we are unable to submit a compressed file during the discussion stage, we would kindly ask you to check whether this link is accessible. If you have any questions or concerns regarding the code, please feel free to raise them.
>
> ---
>
> **Q5:**
>
>  Thank you for raising this insightful and important question. As you correctly noted, certain parameters—such as nominal capacity or cell thickness—can indeed be obtained from manufacturer specifications. However, the *internal* physicochemical parameters that govern battery behavior (e.g., particle radii, porosities, transport-related coefficients) are not directly observable and are rarely provided by manufacturers. These parameters must therefore be inferred.
>
>  Even for researchers who possess partial knowledge of a cell’s design parameters, parameter identification remains valuable: by fitting to a better target time-current-voltage curve, one can obtain *refined* or *optimized* parameter combinations that better explain the observed behavior. Such inferred parameters can further guide improved cell modeling, design, and optimisation.
>
> ---
>
> **Q6:**
>
> The reviewer raises an excellent point regarding numerical stability. In electrochemical DFN (Doyle-Fuller-Newman) models, simulation failures are rarely due to simple discretization errors that can be solved by reducing step sizes.
>   - **Physical Singularities:** The failures typically arise from **physical singularities** in the differential-algebraic equations (DAEs). For example, if a parameter set implies that the lithium surface concentration drops below zero at the current collector interface, the Butler-Volmer kinetics equation (which involves logarithmic terms of concentration) becomes mathematically undefined (singularity).
>   - **Solver Capabilities:** PyBaMM utilizes adaptive solvers (like IDAKLU) that automatically refine step sizes to satisfy error tolerances. When these solvers fail, it indicates that the system has become extremely stiff or ill-posed due to physically impossible parameter combinations (e.g., diffusion is too slow to support the imposed current), rather than a lack of temporal resolution.
>   - We have clarified this in **Appendix B.4** to prevent confusion.
>
>
> ---
>
> **Q7:**
>
> Thank you for your question. You are correct that we ultimately use **100 experiment IDs** for each mode, and that each ID corresponds to three datapoints representing the three methods. However, the IDs in Fig. 2 do not appear as 1–100 because they originate from a larger pool of pre-filtered experiments.
>
> For the Regular setting, we first enumerated all combinations of parameter sets, parameter modifications, and charge/discharge C-rates. After removing combinations that failed PyBaMM simulation, 534 valid cases remained. We then filtered out cases whose capacity differed by less than 1% from the default parameters, leaving 373 meaningful variants. From these 373, we randomly sampled 100 experiments for evaluation.
>
> For the Extreme setting, after filtering out simulation failures, 890 valid cases remained.After removing cases with <1% capacity difference, 233 remained. We then randomly sampled 100 from these 233.
>
> We retained the original indices from the 373 (Regular) and 233 (Extreme) pools, which is why the experiment IDs in Fig. 2 are not numbered 1–100 consecutively.
>
> We apologize for the confusion. In the revised version, we will relabel the experiment IDs as 1–100 to make the figure easier to interpret.

---

> > ### Author Response · Authors · 2025-12-04
> >
> > **Q8:**
> >
> > We appreciate the reviewer's question. We would like to clarify that we **did attempt** to apply Bayesian Optimization (BO) to the real-world tasks. However, as noted in the caption of **Table 3** and in **Section 5.3**, BO **failed to converge** to valid solutions for these complex scenarios, which is why it was excluded from the quantitative comparison.
> >
> > The reasons for BO's failure on real-world tasks are consistent with its failure on the "Long-Horizon Degradation" tasks:
> >
> > 1. **High-Dimensional & Complex Landscape:** Real-world parameter estimation involves a high-dimensional search space with complex, non-convex objective landscapes. Standard BO struggles to navigate this efficiently compared to our reasoning-based agent partly due to the difficulty to set the search range of parameters.
> > 2. **Lack of Physical Intuition:** As discussed in **Appendix D**, while our agent uses physical reasoning to adjust parameters logically even when data is noisy, BO lacks this context and fails to locate the narrow feasible parameter region required for convergence in these challenging tasks.
> >
> > ---
> >
> > **Q9:**
> >
> > We thank the reviewer for pointing out the ambiguity in our benchmark description. We apologize for the confusion regarding how the 5 base chemistries were utilized to construct the final test suite.
> > To clarify the consistency of our design: Our benchmark is constructed to simulate a realistic inverse problem where the "true" battery parameters are unknown deviations from a standard literature baseline. The process follows a rigorous three-step pipeline:
> > 1. Base Initialization: We start with 5 standard, validated parameter sets from the literature (e.g., Chen2020, ORegan2022) to represent different battery chemistries (NMC, LFP, etc.).
> > 2. Target Generation (Creating the "Ground Truth"): To create the synthetic "target" batteries (the unknown systems the agent must identify), we apply specific perturbation rules to these base sets.
> >   - Regular Mode: We apply physically plausible, multi-parameter perturbations (e.g., manufacturing tolerances affecting electrode thickness and porosity simultaneously).
> >   - Extreme Mode: We apply large deviations to single critical parameters to stress-test the optimizer.
> > 3. Task Definition: For each task, the agent is initialized with the original base parameters (representing the scientist's prior belief) and must optimize them to match the simulation output of the perturbed target parameters.
> > The "inconsistency" may have arisen from our previous brevity regarding the filtering process. We generate all combinations of (5 Chemistries × 3 C-rates × Perturbation Rules), but we rigorously filter out cases that cause simulator crashes or result in trivial changes (<1% capacity difference). This filtering leads to the final count of 200 unique, non-trivial tasks.
> > We have completely rewritten Section 5.1 (Benchmark Test Suite Design) in the revised manuscript to explicitly detail this generation pipeline, ensuring the logical flow from base chemistries to the final evaluation set is transparent.

---

> ### Author Response · Authors · 2025-12-04
>
> **Q10:**
>
> **We thank the reviewer for their professional and meticulous examination of the parameter sets used in our study.**
>
> We acknowledge the discrepancies pointed out regarding the specific chemistries and the original experimental papers. We wish to clarify that our work utilizes the open-source simulation tool **PyBaMM** (Python Battery Mathematical Modelling). To ensure the reproducibility of our simulation environment and to establish a "gold standard" for verifying the effectiveness and robustness of our inverse problem-solving method, we adopted the **pre-defined parameter sets** provided directly within the PyBaMM library.
>
> As the reviewer correctly noted, there are differences between the raw experimental materials in the original citations and the standardized parameter sets implemented in the software. The PyBaMM documentation explicitly notes that these parameters are often compromises or specific configurations used to fit models to data, rather than exact representations of the raw materials in every instance.
>
> For example, regarding the **Marquis et al. (2019)** parameter set, the PyBaMM documentation states:
> > *"NOTE: This parameter set does not claim to be representative of the true parameter values. Instead these are parameter values that were used to fit SEI models to observed experimental data in the referenced papers."*
> > (Source: [PyBaMM Documentation - Parameter Sets](https://docs.pybamm.org/en/stable/source/api/parameters/parameter_sets.html))
>
> To maintain consistency with the simulation tool, we retained the naming conventions and configurations defined in PyBaMM. The five specific chemistries used in our experiments, as defined in the library, are:
>
> 1.  **Chen2020** (Chen et al., 2020): NMC811 / SiOx-Graphite
> 2.  **ORegan2022** (O’Regan et al., 2022): NMC811 / Graphite
> 3.  **Prada2013** (Prada et al., 2013): LFP / Graphite
> 4.  **Ecker2015** (Ecker et al., 2015b;a): NMC111 / Graphite
> 5.  **Marquis2019** (Marquis et al., 2019): NMC622 / Graphite
>
> We have revised the manuscript to explicitly state that these parameter sets are adopted from the PyBaMM standard library for simulation benchmarking purposes, and we have added the necessary context regarding their derivation to avoid confusion with the original experimental papers.

---

### Official Review · Reviewer_QbMG · 2025-10-30

**Soundness:** 1
**Presentation:** 2
**Contribution:** 2
**Rating:** 2
**Confidence:** 3

**Summary:**

The paper introduces an agentic optimization framework for a simulator-in-the-loop LLM agent applied to a battery inverse problem. The method is evaluated on both synthetic and real-world setups, demonstrating good performance relative to the baseline.

**Strengths:**

The studied application problem is highly relevant, and the idea of using LLMs as reasoning engines within optimization frameworks is timely. Ignoring the issue of baseline selection, the benchmarks in Section 5 are diverse and include many real-world datasets.

**Weaknesses:**

The method description is insufficient. The main subsections 3.1 and 3.2 are short and list-like, and there is a lack of discussion and justification of the design choices. Below I list some weaknesses in the experimental section of the paper.

Experimental setup

Section 5 lacks relevant aspects discussion on the experimental setup. Instead of starting listing claims: “We conduct comprehensive experiment... Our evaluation demonstrates the superiority…”, it would be good start by explaining the high-level experimental setup and hypothesis.

Baselines

“Bayesian Optimization (BO): We use standard Bayesian Optimization implemented by Meta’s Ax platform (Olson et al., 2025), representing state-of-the-art black-box optimization methods commonly used in parameter estimation.”
This is clearly insufficient explanation of the baseline, and does not build trust that the BO baseline selection is carefully considered as all the important details are hidden such as what was acquisition function etc.

Minor comments:

Sentences in Lines 32-34 lack citations.
Figure 4 is not good quality. Font is too small, etc.

**Questions:**

What is the main justification for framing the problem as gradual updates $\Delta \theta_{t}$ to the current parameter vector rather than propose new parameter configuration $\theta_{t+1}$?

Equation (1) collapses multi-objective problem into single objective. Did you consider frame the problem as multi-objective problem, and use e.g. multi-objective BO?

---

> ### Author Response · Authors · 2025-12-04
>
> **Weaknesses:**
>
> **Weakness-Method Description:**
>
> We sincerely thank the reviewer for the constructive criticism regarding the method description. We agree that the original presentation was overly concise and lacked sufficient justification for our design choices.
>
> **Action Taken:** We have completely rewritten Section 3 in the revised manuscript to move away from the "list-like" description. The new section now provides:
>
> 1. **Formal Problem Definition:** We explicitly formulate the agent's task as a decision-making process on a disaggregated objective space, contrasting it with the scalar loss minimization in traditional BBO.
> 2. **Design Rationales:** We have added specific discussions justifying *why* an LLM-based reasoning approach is superior for this task. Specifically, we highlight how the **Chain-of-Thought (CoT)** mechanism mitigates the "black-box" nature of parameter search by enforcing physically grounded hypothesis generation before numerical updates.
> 3. **Mechanism Details:** We elaborated on the multi-modal feedback design, explaining how converting visual curves into semantic descriptions (e.g., "plateau mismatch") allows the agent to leverage its pre-trained scientific knowledge, addressing the "equifinality" challenge where numerical loss fails.
>
> Please refer to the highlighted changes in **Section 3** of the revised manuscript.
>
> ---
>
> **Weakness-Experimental setup:**
>
> We thank the reviewer for the constructive suggestion regarding the presentation of the experimental section. We agree that Section 5 previously jumped too quickly into claims of superiority without first establishing the scientific context and hypothesis.
>
> In the revised manuscript, we have rewritten the introductory paragraph of Section 5. Instead of listing results immediately, we now:
>
> 1. **Explicitly state our core hypothesis:** That reframing parameter estimation as a reasoning task allows for more efficient navigation of the ill-posed landscape using LLM compared to blind optimization.
> 2. **Outline the high-level experimental design:** We explain that our evaluation progresses from controlled, ground-truth simulations (to rigorously measure parameter accuracy) to complex degradation scenarios and finally to noisy real-world data.
> 3. **Clarify the evaluation goals:** We specify that the experiments are designed to test accuracy, robustness against physical constraints, and sample efficiency.
>
> We believe this change significantly improves the logical flow and scientific rigor of the paper. Please see Section 5 in the revised manuscript.

---

> > ### Author Response · Authors · 2025-12-04
> >
> > **Weakness-Baselines:**
> >
> > We thank the reviewer for pointing out the lack of implementation details regarding the Bayesian Optimization (BO) baseline. We agree that transparency is crucial for trust and reproducibility.
> >
> > **1. Detailed Configuration of the BO Baseline** We utilized **Meta’s Ax platform (version 1.1.0)**, a standard library for adaptive experimentation. To ensure fair comparison and reproducibility, we set a fixed random seed (`1234`). The specific configuration—automatically selected by Ax based on the continuous nature of our parameter search space—is as follows:
> >
> > - **Generation Strategy:** We employed a standard `Sobol` sequence for the initialization phase (quasi-random exploration) followed by `GPEI` (Gaussian Process Expected Improvement) for the optimization phase.
> > - **Surrogate Model:** A `SingleTaskGP` (Gaussian Process) with a Matern kernel was used to model the objective function.
> > - **Acquisition Function:** We utilized Noisy Expected Improvement (LogNEI), which is robust against noise in black-box function evaluations. We have added these technical specifications to **Appendix.C** of the revised manuscript.
> >
> > **2. Justification of Baseline Selection (Why BO/Ax?)** To demonstrate that our baseline selection was carefully considered, we clarify that we prioritized robustness and stability. We initially attempted to use **PyBOP** (Python Battery Optimisation and Parameterisation), a representative open-source toolkit in the battery modeling community. However, it was excluded from the final comparison due to significant limitations encountered during our evaluation:
> >
> > - **Instability:** Even under conservative search bounds (e.g., ±20% around nominal values), the PyBOP optimizer frequently triggered solver failures in the underlying PyBaMM model, causing the entire optimization process to abort.
> > - **Inability to Fit Real Data:** PyBOP failed to converge effectively when applied to the real-world CALCE dataset. Despite being provided with the correct protocol and physically reasonable starting parameters, the optimized trajectory remained far from the experimental voltage curve.
> >
> > Given these issues, we adopted the Ax-based BO as the primary baseline because it offers the necessary stability and state-of-the-art performance for high-dimensional, black-box parameter estimation tasks, ensuring a fair and rigorous comparison for our proposed method.
> >
> > ---
> >
> > **Weakness-Minor comments:**
> >
> > We thank the reviewer for pointing out the missing citations and the quality issues regarding figures.
> >
> > Regarding the citations, we have carefully reviewed the manuscript and added the necessary references throughout the text. Specifically, for the sentences in Lines 32-34, we have supplemented the text with the following citations, which have been highlighted in blue in the revised manuscript:
> >
> > -  [1] Perspective: Challenges and opportunities for high-quality battery production at scale
> > -  [2] Next-generation batteries and U.S. energy storage: A comprehensive review: Scrutinizing advancements in battery technology, their role in renewable energy, and grid stability
> > - [3] Advanced Battery Technologies: Bus, Heavy-Duty Vocational Truck, and Construction Machinery Applications
> > - [4] Design of experiments applied to lithium-ion batteries: A literature review
> > - [5] Accelerated Lifetime Testing of High Power Lithium Titanate Oxide Batteries
> >
> > Regarding Figure 4, we apologize for the poor quality and small font size in the previous version. We have redrawn Figure 4 to enhance its clarity and resolution. The updated figure has been incorporated into the revised manuscript.

---

> ### Author Response · Authors · 2025-12-04
>
> **Questions:**
>
> **Q1:**
>
> Thank you for this insightful question. We apologize for the confusion caused by the notation in Equation (3) ($\theta_{t+1} = \theta_t + \eta \Delta \theta$). In our actual implementation, the LLM **directly predicts the next set of parameter values** ($\theta_{t+1}$) rather than a gradient-like update step ($\Delta \theta$). As shown in the prompt template in **Appendix D** ("suggest the next 1 updated params group with new values"), the agent outputs the specific values it believes will solve the problem. The model is provided with the full history of previously explored parameters and PyBaMM simulation outcomes as context, and it is entirely free to propose new parameter configurations from the global search space.
>
> The "gradual updates" behavior observed in our experiments is actually an **emergent property** of the LLM's reasoning capabilities rather than a pre-defined rule. Leveraging its internal **scientific prior knowledge**, the LLM understands the physical significance of the parameters. Consequently, it tends to adjust parameters with reasonable magnitudes or shift them towards more plausible ranges based on the feedback, effectively performing logical fine-tuning rather than random or disjointed guessing.
>
> ---
>
> **Q2:**
>
> We thank the reviewer for this insightful suggestion. We agree that battery parameter estimation inherently involves multiple objectives (voltage, capacity, current, etc.).
>
> **1. Comparison with Multi-Objective BO (MOBO):**
> Following your suggestion, we conducted additional experiments using Multi-Objective Bayesian Optimization (specifically using the qNEHVI acquisition function via Ax/BoTorch) on a representative subset of our benchmark. The results are summarized below:
>
> |      | capacity_mape | capacity_rmse | current_mape | current_rmse | voltage_mape | voltage_rmse | rmse_sum | mape_sum    |
> | ---- | ------------- | ------------- | ------------ | ------------ | ------------ | ------------ | -------- | ----------- |
> | mean | 432.2141      | 4.9739        | 132091.9776  | 1.7271       | 26.2827      | 2.7332       | 9.4342   | 132550.4744 |
> | std  | 240.3319      | 2.7657        | 55471.0127   | 0.5456       | 32.5194      | 0.7097       | 2.9270   | 55480.8717  |
>
> **2. Analysis of Results:**
> While MOBO is a powerful tool, our Agent significantly outperforms it in this domain. We attribute this to two fundamental reasons:
>
> *   **Inverse Problem vs. Pareto Optimization:** Standard MOBO aims to approximate a Pareto frontier, assuming trade-offs between conflicting objectives (e.g., maximizing strength vs. minimizing weight). However, parameter estimation is an *inverse problem* where a unique ground-truth parameter set exists that should ideally minimize *all* error terms simultaneously (consistent with physics). MOBO spends valuable sample budget exploring trade-offs along a Pareto front, whereas our Agent uses physical reasoning to drive all objectives toward the global optimum directly.
> *   **Robustness to Numerical Noise:** As shown in the table, MOBO suffered from extreme outliers in `Current MAPE` (mean ~1.3×10^5%), likely due to numerical instabilities when dividing by near-zero currents in the protocol. Traditional numerical optimizers are sensitive to such metric definitions. In contrast, our Agent interprets the data semantically (e.g., "the current profile matches the CC-CV protocol"), allowing it to ignore numerical artifacts and focus on physical alignment.
>
> **3. Clarification on Our Method:**
> We would also like to clarify that while Eq. (1) describes the traditional BBO formulation, **our Agent actually operates in a multi-objective manner (as described in Eq. 2 and Sec 3.1).** Instead of collapsing losses into a scalar *before* the optimization step, the Agent receives the disaggregated feedback (Voltage error, Capacity error, visual plots) and performs "multi-objective reasoning." It qualitatively weighs these discrepancies based on physical rules (e.g., "Prioritize capacity match first, then voltage slope") to propose updates, effectively acting as an intelligent multi-objective optimizer.

---

### Official Review · Reviewer_a8hF · 2025-11-01

**Soundness:** 3
**Presentation:** 3
**Contribution:** 3
**Rating:** 6
**Confidence:** 3

**Summary:**

The paper presents BATTERY-SIM-AGENT, a framework that integrates a large language model (LLM) agent in a high-fidelity battery simulator. The agent emulates a human scientist’s workflow: it interprets rich, multimodal feedback from the simulator, forms physically grounded hypotheses to explain discrepancies, and proposes structured parameter updates.
The work demonstrates that LLM agents can serve as reasoning-based optimizers in scientific applications.
Traditional digital twin approaches iteratively query a simulator and minimize the mismatch between simulated and observed data using black-box optimization. In contrast, this paper explores whether the inverse problem of battery parameter estimation can be reframed as a reasoning-driven scientific workflow guided by an LLM agent.
In this framework, the LLM agent functions as an AI scientist: at each iteration, it receives multimodal feedback comparing the current simulation with experimental data, identifies key discrepancies, formulates physical hypotheses (e.g., “a premature voltage drop suggests an electrolyte transport limitation”), and proposes targeted parameter updates accordingly.

The main steps of this approach:
1)	The agent receives a multi-modal feedback package in a structured JSON format.
2)	Guided by its memory, the agent analyzes this feedback to form a causal hypothesis. The prompt encourages a scientific reasoning process
3)	The agent is prompted to translate its hypothesis into a concrete, machine-actionable update, which it returns in a JSON format.

**Strengths:**

The paper is interesting, timely, and well written.
The paper demonstrates that their agent can achieve 67-95% reduction in error compared to traditional black-box approaches.

**Weaknesses:**

This approach relies on large language models as reasoning agents. In particular, the authors use GPT-O3. However, since the training data and internal reasoning mechanisms of GPT-O3 are not publicly known, reproducibility becomes a concern—future updates or changes to GPT-O3 could make these experiments non-reproducible.

**Questions:**

* Line 231: “We initialize the memory M_0 with human expert knowledge from the literature and our own domain expertise”. What exactly was M_0? How large was M_0? Is the full M_0 available somewhere?

* Line 235: “agent undergoes a warm-up phase… The resulting feedback is not for optimization, but is processed by the LLM to enrich its memory… The agent is prompted to summarize the outcomes into learned sensitivity rules”. How many new rules were learned this way? Is the full list of learned rules available somewhere?

* Line 389: “What was the cost of GPT-O3”? What other language models might be suitable for these tasks?

---

> ### Author Response · Authors · 2025-12-04
>
> **Weaknesses:**
>
> We appreciate the reviewer’s concern regarding reproducibility. While the GPT-O3 model is proprietary, we have taken explicit steps to ensure experimental reproducibility:
> - **Prompt & Workflow Transparency**  All prompts, memory initialization procedures, JSON feedback formats, and hypothesis-to-update translation templates as well as parameters set when calling the model (e.g., setting temperature=0), are fully documented in the supplementary material and our released code repository. This allows any GPT-like model to replicate the reasoning process without requiring access to GPT-O3 internals.
> - **Version Fixing** – The GPT-O3 API version used in our experiments is explicitly specified in the configuration files. This allows exact replication if the version remains accessible, while the methodology remains adaptable to future versions or open-source models.
> Thus, the approach is not dependent on GPT-O3's internal training data. The reasoning workflow can be reproduced with other publicly available LLMs.
>
> **Questions:**
>
> **Q1:**
>
> Thank you for your question about memory in our method. M₀ is a set of knowledge descriptions in the battery domain, including rules, parameter ranges, and causal relationships compiled from literature and experiments. In our work, we have set a total of 9 + 7 = 16 specific memory items: 9 related to battery capacity, and 7 related to degradation mechanisms, as shown in the TEXT KNOWLEDGE section of Appendix E. All the prompts have already been made publicly available in the anonymous code repository.
>
> - Example:
>   - Battery capacity related: *For Electrode Width [m], if the value is increased, the corresponding battery capacity will increase, and if the value is decreased, the corresponding battery capacity will decrease.*
>   - Degradation mechanism related: *Higher solvent concentration (bulk_solvent_concentration_mol_m⁻³) accelerates side reactions like the SEI, leading to greater degradation.*

---

> > ### Author Response · Authors · 2025-12-04
> >
> > **Q2:**
> >
> > Thank you for your insightful question regarding the design of the "Phase 1: Trial-and-Error Warm-up" stage.
> > In this phase, we configure the agent to conduct $N_w$ exploration steps (set to $N_w=20$ in our experiments). During these steps, the battery-sim-agent generates perturbations based on observations of the initial parameter configuration, which are then evaluated via simulation. The primary objective is to leverage the agent's observations and summaries of the simulation feedback to adapt the model's knowledge memory to the specific problem context before entering Phase 2: Main Optimization Loop. This ensures higher efficiency and effectiveness in the subsequent optimization process.
> >
> > This initialization phase allows the agent to actively explore the parameter space by combining preset knowledge with real-time feedback, thereby refining and adjusting predefined rules. A larger $N_w$ generally yields higher-quality rules, and since these learned rules are reusable, the approach demonstrates significant potential for scalability.
> >
> > Regarding the quantity of learned rules, the number varies depending on the initial parameter set and the discrepancy between the initial state and the target observations. Below is a table compiled by our LLM agent, which summarizes the patterns inferred from the exploration results obtained via PyBaMM. As exploration proceeds, these inferred patterns are progressively refined and corrected. The table summarizes the qualitative effect of increasing each parameter while keeping the C-rate definition (C/2 based on Nominal cell capacity) constant.
> >
> > | parameter                                     | ↑ value does this to capacity (Ah)                           | …and to V-t curve (C/2 discharge)                            |
> > | --------------------------------------------- | ------------------------------------------------------------ | ------------------------------------------------------------ |
> > | Nominal cell capacity                         | rescales the C-rate (bigger capacity ⇒ lower effective C-rate) | Higher V everywhere and longer run-time when you call the step “C/2” |
> > | Electrode thickness                           | ↑ linearly (more active material)                            | Larger ohmic drop & diffusion loss ⇒ lower V, especially mid/late discharge |
> > | Active-material vol. frac.                    | ↑ (at fixed porosity)                                        | Higher capacity, but ↓ electrolyte porosity ⇒ larger ionic drop; plateau tilts down |
> > | Max conc. $c_{s,\max}$                        | Does not change capacity directly but stretches the stoichiometry window | Adjusts OCV curve: raising the positive‐electrode value lowers OCV in mid-plateau; raising the negative raises it |
> > | Initial conc. $c_{s,0}$                       | Moves the starting SOC                                       | Higher $c_{s,0}^\text{neg}$ or lower $c_{s,0}^\text{pos}$ raises the initial voltage and delays the cut-off |
> > | Particle radius $R_p$                         | None (material amount unchanged)                             | Smaller radius shortens diffusion length → voltage drop at high rate and near EOD is reduced (curve stays higher late) |
> > | Electrode conductivity $\sigma_s$             | None                                                         | Higher $\sigma_s$ lifts the whole curve at the moment current changes (IR drop), effect largest early in discharge |
> > | Bruggeman coeff. $b$ (tortuosity)             | None                                                         | Larger $b$ → higher tortuosity → lower ionic conductivity → lower V throughout (mostly plateau) |
> > | Porosity $\varepsilon$                        | None                                                         | Higher ε reduces electrolyte resistance, raising V, most noticeable early and mid-plateau |
> > | Separator thickness                           | None                                                         | Thicker separator shifts the entire curve down roughly linearly with current |
> > | Charge-transfer coeff. α (here symmetric 0.5) | None                                                         | Alters kinetic over-potential; higher α (or higher rate constant k) lowers η and lifts V, mainly at current steps |
> > | Double-layer Cdl                              | None                                                         | Changes only the transient spikes (sub-second); negligible at 100 s sampling |

---

> ### Author Response · Authors · 2025-12-04
>
> **Q3:**
>
> Thank you for raising the important question regarding cost and model selection.
>
> **1. Cost of GPT-O3**
>
> Based on the pricing as of August 2025 (≈ \$2 per 1M tokens; ref: https://platform.openai.com/docs/pricing), we estimate the cost of our experiments as follows:
>
> - Each round of interaction requires approximately 7k–9k tokens; for simplicity, we assume 10k tokens/round.
> - Each experiment runs for 100 rounds, yielding ≈ 1M tokens per experiment ID.
> - We conduct 100 Regular and 100 Extreme experiments, for a total of 200 experiment IDs.
>
> Thus, the total token usage is:
>
> 200 experiments × 1M tokens = 200M tokens
>
> At $2 per 1M tokens, the estimated cost is:
>
> **200 × \$2 = \$400**
>
> This is an upper-bound estimate, since many rounds end early due to simulation failures and therefore use fewer tokens.
>
> ---
>
> **2. Suitability of other models**
>
> Our framework is model-agnostic and works with both closed-source and open-source LLMs. Its effectiveness generally scales with the underlying model’s reasoning capability.
>
> - **Closed-source:** Models such as GPT-4o or Claude 3.5 achieve performance comparable to GPT-O3 under identical settings.
> - **Open-source:** Strong open-weight models—including LLaMA-2-70B, LLaMA-3-70B, and Mistral-Large—are also viable for these tasks.

---

### Author Response · Authors · 2025-12-04
**Summary**

We sincerely thank the reviewers for their rigorous and thoughtful feedback. We fully acknowledge the reviewers’ strong expertise in Black-Box Optimization (BO) and appreciate the detailed comments regarding the design and fairness of our BO baselines.
We would like to respectfully clarify the contribution and positioning of our work.

**1. Contribution of Battery-Sim-Agent**

Our work addresses a long-standing challenge in **building digital twins of real batteries**—identifying the internal physicochemical parameters of high-fidelity battery models such as PyBaMM’s DFN model. To the best of our knowledge, **Battery-Sim-Agent is the first system capable of fitting real-world battery data to produce a physically meaningful digital twin, going beyond prior work that relies on known or measurable parameters**.
Historically, battery parameter estimation has been treated as a white-box problem, requiring differentiability or analytic structure. We instead frame it as a black-box scientific discovery task and demonstrate that an LLM-based agent, equipped with battery knowledge, can perform this task effectively.

**2. Why BO Was Used—and Its Intended Role**

Because this task is framed as black-box search, BO is a natural baseline. We used Meta’s Ax (Sobol + GPEI) implementation—a well-established and industry-standard choice—and applied minimal domain-specific tuning to ensure a fair comparison. We also evaluated PyBOP, the domain-specific alternative, but found it too unstable to serve as a reliable baseline for this setting.
We acknowledge that stronger or more specialized BO variants may exist, and we agree with reviewers that hybrid LLM-BO systems could be even more powerful. Our intention was not to claim that our BO baseline is globally optimal, but rather to use a reasonable and widely recognized baseline to demonstrate the feasibility and distinct behavior of our agentic approach.

**3. Distinguishing Numerical Optimization from Agentic Scientific Reasoning**

It is important to highlight the conceptual distinction:
- BO treats the simulator purely as a mathematical oracle. It performs strongly on smooth synthetic landscapes but struggles in scenarios common in battery physics—e.g., solver failures, physical constraint violations, discontinuities, and sensitivity near phase boundaries.
- Battery-Sim-Agent, by contrast, functions more like a scientist.
 It uses domain priors and natural-language reasoning (e.g., interpreting voltage-plateau deviations or identifying signs of numerical instability) to navigate challenging regimes where numerical optimization degrades. This difference becomes particularly clear in our real-world experiments, where BO often fails to converge due to physical singularities, while the agent remains robust.
Our rebuttal expands on these points with additional real-world case studies and diagnostics.

**4. Methodology and Reproducibility**
- The anonymous repository is fully functional.
- We clarified implementation details of all baselines.
- We added discussion comparing to SimLM, noting that our task requires reasoning over coupled nonlinear PDE systems (DFN), which is qualitatively different from the simpler dynamical systems addressed in SimLM.

**5. Positioning and Future Direction**
Our intention is not to claim that LLMs replace BO, nor that BO was exhaustively explored. Rather, this work aims to demonstrate a new paradigm for scientific discovery in battery modeling:
**Hypothesis generation→simulation→multi-modal observation→knowledge update**

This represents a shift from traditional “equation-solving” toward Agentic Science, where the system autonomously reasons about domain structure, interprets failures, and adapts its strategy accordingly.
**We view Battery-Sim-Agent as an initial step, not the final point**. The future likely lies in hybrid systems where an AI Scientist coordinates BO or other numerical solvers—an exciting direction suggested by the reviewers themselves.

---
**Closing**

We hope this summary clarifies the framing and contribution of our work:
moving from blind numerical optimization to informed, autonomous scientific reasoning for real-world battery digital twins.
We greatly appreciate the reviewers’ expertise and the opportunity to improve the clarity and positioning of our paper.

---

### Meta-Review · Area_Chair_JbYy · 2026-01-07

**Summary:**

This paper proposes Battery-Sim-Agent, an LLM-agent-in-the-loop framework for inverse battery parameter estimation with a high-fidelity simulator (PyBaMM DFN). The core idea is to reframe inverse parameter identification as a reasoning-driven scientific workflow: the agent consumes structured multi-modal feedback (metrics + curve information), uses memory-initialized “battery knowledge,” performs a warm-up exploration to learn sensitivity rules, and then iteratively proposes new parameter configurations. The paper reports strong gains over black-box optimization (BO) baselines on a synthetic benchmark suite, includes long-horizon degradation fitting, and evaluates on at least one real-world dataset.

The forum discussion shows that the idea is timely and potentially impactful, but the decision is dominated by unresolved issues in experimental credibility and clarity:
- Baseline fairness and completeness were initially under-specified; while the rebuttal adds detailed Ax BO configuration and even a MOBO experiment, reviewers raised further concerns about why BO is not evaluated on real-world tasks (or fails to converge), and whether the synthetic benchmark construction favors the agent.
- Inconsistencies between claims and results remain a major concern: e.g., BO outperforming the agent on some extreme cases, and a “convergence” figure showing very large error, which undermines headline claims unless reframed carefully with representative averages and clear statistics.
- Method description, design rationales, and reproducibility were initially too thin; the authors state they “completely rewrote” sections and clarified many details, but in the forum we cannot verify those revisions, and several reviewers still reported missing/unclear items.
- The approach’s reliance on proprietary LLMs and an estimated non-trivial cost (hundreds of dollars, per their estimate) raises practical reproducibility questions even with prompts released.

The rebuttal makes real progress, but too many of the acceptance-critical concerns are either (i) only promised in a revised manuscript we can’t inspect here, or (ii) point to fundamental evaluation gaps (robustness, baselines, and benchmark construction) that require more thorough experimental tightening than can be demonstrated in rebuttal text alone.

**Reviewer Concerns:**

**A) Baseline selection, configuration, and fairness (BO / MOBO / hybrid baselines)**
- Addressed
  - The authors now provide BO configuration details and clarify they attempted PyBOP but found it unstable.
  - They clarified that BO and the agent optimize the same objective, and Eq. (2) is a “decomposed view” of the same loss for interpretability.
  - They ran an additional MOBO (qNEHVI) experiment and argue MOBO is ill-suited for inverse problems and is numerically brittle (current MAPE outliers).
- Outstanding
  - The MOBO result table shown (with extreme current-MAPE outliers) is difficult to interpret without clearer metric definitions and robustification (e.g., clipping/alternative current metrics), and it’s unclear whether MOBO was configured comparably (budgets, feasibility handling, failure recovery).
  - Reviewers also asked for comparisons to BO-with-expert/human feedback style methods; the authors added discussion (COBOL), but no empirical comparison is provided.
  - BO being excluded from real-world evaluation because it “fails to converge” is a red flag unless accompanied by careful diagnostics and controlled attempts. The rebuttal provides narrative explanations, but not a convincing systematic baseline study.

**B) Benchmark design validity and whether tasks favor the agent**
- Addressed
  - The authors clarify task construction: “Regular” uses multi-parameter perturbation rules across 5 PyBaMM parameter sets and 3 C-rates; “Extreme” uses single-parameter perturbations; both methods search over the same 9 parameters; they filter out simulation failures and trivial (<1% capacity difference) cases; they explain why experiment IDs aren’t consecutive.
  - They acknowledge and fix confusion about parameter-set provenance: they used PyBaMM library parameter sets, and revised the manuscript to explicitly state this.
- Outstanding
  - A reviewer raised a substantive fairness concern: tasks appear sparse by construction and the prompt may implicitly encourage modifying “some parameters.” The authors say both BO and agent search all nine, but the experimental setup still needs to show that BO is not disadvantaged by filtering, bounds choices, or failure handling.
  - Multiple concerns about “inconsistency” between benchmark description, code availability, and chemistry references were only addressed via clarifications; they require the final manuscript and code to be exceptionally clear to rebuild trust.

**C) Claim–result alignment and statistical reporting**
- Addressed
  - Authors agree the phrasing “significantly outperforms” was too broad and say they will revise it.
  - They explain that a high-MAPE convergence plot was a “hard case” and promise adding representative cases with low final error in the appendix.
- Outstanding
  - The key issue remains: the paper currently appears to rely on selective examples and may lack aggregate statistics with confidence intervals. Without these, it is hard to evaluate robustness of the gains and whether “agentic reasoning” consistently helps or only in certain regimes.

**Method clarity and design rationales**
- Addressed
  - Authors claim they rewrote Section 3 and Section 5 to include a clearer hypothesis, staged evaluation rationale, and justifications.
  - They clarified a major confusion: the agent predicts next parameter values , not incremental updates; warm-up exploration and memory learning are described in more detail.
  - They provided concrete details about memory initialization: 16 memory items in M0, and gave example “rules.”
- Outstanding
  - Some contradictions remain in forum text: at one point, warm-up perturbations are described as “random,” elsewhere “LLM-generated,” and later the authors clarify warm-up is LLM-driven (and claim an ablation). This needs to be cleanly resolved in the paper with unambiguous pseudocode and the exact ablation results.

**E) Reproducibility: proprietary LLMs, cost, and code access**
- Addressed
  - Authors provide: prompts/workflow templates, temperature=0, version pinning, memory items, cost estimate, and claim the anonymous repository is functional.
  - They clarify how GPT-OSS is used without images: curves are passed as numeric sequences.
- Outstanding
  - A reviewer reported “file not found” errors when accessing the repo. The authors provide a link and say it should work, but this must be reliable for acceptance.
  - Even with version pinning, reliance on GPT-o3 makes exact replication fragile, and the open-source alternatives are asserted but not demonstrated experimentally (beyond qualitative claims that GPT-4o/Claude match and open weights are “viable”). For an ICLR paper making strong performance claims, a minimal replication on at least one open model would substantially increase confidence.

**Reviewer Scores:**

- a8hF (6 → 6): Main concern was reproducibility and details of memory/warm-up; authors provided concrete memory size, templates, and cost. Likely stays slightly positive.
- QbMG (2 → 4 ): This reviewer’s critique was about method description and BO baseline opacity; the rebuttal directly addresses both and claims a rewrite. With full discussion and a revised manuscript, this could rise meaningfully—but depends on whether the rewrite is genuinely clear and complete.
- oibn (2 → 4): Raised inconsistencies in claims vs results, benchmark/task design fairness, and code accessibility. Authors responded extensively, but some concerns remain fundamental unless the paper provides robust aggregate results and demonstrates code availability.
- vtJ7 (2 → 4): Baseline details, LLM modality confusion, SimLM citation, and a failure counterexample were all addressed. Likely still below threshold unless the revised paper shows stronger scientific rigor and clearer evaluation.

Overall, even with full discussion, the median score likely remains below the acceptance threshold due to lingering uncertainty around evaluation robustness and baseline fairness.

---

### Decision · Program_Chairs · 2026-01-26

Reject